# Satb2 acts as a gatekeeper for major developmental transitions during early vertebrate embryogenesis

Saurabh J. Pradhan [1], Puli Chandramouli Reddy [1], Michael Smutny[2,3], Ankita Sharma [1], Keisuke Sako [2], Meghana S. Oak[4], Rini Shah[1], Mrinmoy Pal [1], Ojas Deshpande [1,5], Greg Dsilva [1], Yin Tang[6], Rakesh Mishra [7], Girish Deshpande[8], Antonio J. Giraldez [6], Mahendra Sonawane[5], Carl-Philipp Heisenberg [2,10✉] & Sanjeev Galande [1,9,10✉]

Zygotic genome activation (ZGA) initiates regionalized transcription underlying distinct cellular identities. ZGA is dependent upon dynamic chromatin architecture sculpted by conserved DNA-binding proteins. However, the direct mechanistic link between the onset of ZGA and the tissue-specific transcription remains unclear. Here, we have addressed the involvement of chromatin organizer Satb2 in orchestrating both processes during zebrafish embryogenesis. Integrative analysis of transcriptome, genome-wide occupancy and chromatin accessibility reveals contrasting molecular activities of maternally deposited and zygotically synthesized Satb2. Maternal Satb2 prevents premature transcription of zygotic genes by influencing the interplay between the pluripotency factors. By contrast, zygotic Satb2 activates transcription of the same group of genes during neural crest development and organogenesis. Thus, our comparative analysis of maternal versus zygotic function of Satb2 underscores how these antithetical activities are temporally coordinated and functionally implemented highlighting the evolutionary implications of the biphasic and bimodal regulation of landmark developmental transitions by a single determinant.

[1] Centre of Excellence in Epigenetics, Department of Biology, Indian Institute of Science Education and Research, Pune 411008, India. [2] Institute of Science and Technology Austria, Klosterneuburg, Austria. [3] Centre for Mechanochemical Cell Biology and Division of Biomedical Sciences, Warwick Medical School, University of Warwick, Coventry CV47AL, UK. [4] Indian Institute of Science Education and Research, Kolkata 741246, India. [5] Department of Biological Sciences, Tata Institute of Fundamental Research, Homi Bhabha Road, Mumbai 400 005, India. [6] Department of Genetics, Yale University School of Medicine, New Haven, CT, USA. [7] Centre for Cellular and Molecular Biology, Uppal Road, Hyderabad, India. [8] Department of Molecular Biology, Princeton University, Princeton, NJ 08540, USA. [9] Department of Life Sciences, Shiv Nadar University, Gautam Buddha Nagar, Greater Noida, Uttar Pradesh, India. [10] These authors jointly supervised: Carl-Philipp Heisenberg, Sanjeev Galande. ✉email: heisenberg@ist.ac.at; sanjeev@iiserpune.ac.in

The Cambrian explosion resulted in major diversification in the body plans of organisms. Precise body plan determination involves multiple sequentially progressive steps of zygotic genome activation (ZGA), cell-type specification, and organogenesis[1]. During early embryogenesis, transcripts and proteins are either maternally deposited or zygotically synthesized. Maternally deposited factors primarily function during early cleavage cycles of embryogenesis and provide required genome stability by employing post-transcriptional mechanisms of gene regulation, whereas zygotically synthesized factors can regulate gene expression using both transcriptional and post-transcriptional mechanisms[2]. Interestingly, maternally deposited factors classified as 'pioneer factors' exhibit unique ability to bind to closed chromatin and modulate local landscape for global transcription burst necessary for embryonic developmental processes such as patterning and morphogenesis[3–5]. ZGA is the earliest developmental transition at which the pioneer factors impart transcriptional competence to the individual cells by incorporating positional information provided by localized patterning determinants[6]. Previous studies have identified specific regulators of ZGA and the precise mechanisms that control their mode of action to engineer regionalized gene expression have been investigated over the past decade[7–10]. ZGA is actively regulated by employing multiple mechanisms such as increased translation of pioneer factors and dynamic changes in DNA methylation and histone modifications[11,12]. While ZGA is a conserved feature of early embryogenesis and relies upon genome-wide chromatin remodeling to establish transcriptional competence[13,14], how these genome-wide changes are translated into tissue-specific activities of patterning determinants remains largely unclear.

Spatiotemporal gene expression patterns are achieved by employing higher-order Gene Regulatory Networks (GRNs)[15]. GRNs are established via incorporation of novel gene families either de novo or by gene duplication and diversification leading to the acquisition of novel functions[7]. An important protein family which exhibits divergence with the evolution of jawed vertebrates consists of the special AT-rich binding proteins, SATB1 and SATB2. Both these proteins show high structural similarity and harbor a N-terminal ubiquitin-like domain (ULD), two DNA-binding CUT domains, and a C-terminal homeobox domain[16]. Interestingly, SATB2 performs unique functions as a chromatin organizer during embryonic development, tissue specification, and morphogenesis in vertebrates[17,18]. SATB2 binds to multiple regulatory sites in a genome-wide manner and influences chromatin architecture thereby modulating target gene expression[19,20]. In humans and mice, mutations in the SATB2 locus result in cleft palate, craniofacial defects, developmental delay, cognitive disabilities and behavioral abnormalities[21–25]. SATB2 proteins exhibit a high level of conservation across vertebrates including zebrafish[26]. satb2 is maternally deposited and expressed during early stages of zebrafish embryogenesis. Compromising Satb2 function using morpholinos results in severe developmental arrest[27]. Of note, later during organogenesis, loss of BMP and SHH signaling affect satb2 expression in mandibles and pharyngeal arches of mouse and zebrafish embryos respectively[18,28]. However, the molecular circuitry deployed by Satb2 to achieve early cell fate specification leading to primordial organ formation remains unclear.

Towards this, we generated a loss of function mutation in the zebrafish satb2 locus. The mutation faithfully mimics important features of SATB2-associated syndrome in humans including craniofacial abnormalities which often arise from aberrant neural crest (NC) specification and migration[29,30]. Neural crest cells (NCCs) are specialized multipotent cells that contribute to the development of cartilage, pigment cells, bones, and connective tissue[31]. Transcriptome profiling of satb2 mutants revealed global deregulation of genes involved in NC specification and migration. Specifically, our data demonstrate that zygotic Satb2 promotes the expression of the positive regulators underlying NCC specification in zebrafish. Moreover, Chromatin accessibility analysis using Assay for Transposase-Accessible Chromatin with high throughput sequencing (ATAC-seq) performed using satb2 mutant embryos at 14 somites stage suggested changes in global chromatin architecture via remodeling and nucleosome positioning. Genome-wide occupancy analysis of Satb2 in zebrafish and mice highlighted the conserved regulatory mechanisms across the evolutionary scale. Surprisingly, our analysis revealed a previously undocumented function for maternally deposited Satb2. We demonstrate that unlike its zygotic counterpart, the maternally deposited Satb2 functions as a transcriptional repressor of zygotically transcribed genes and controls the timing of ZGA by differentially regulating pluripotency factors. Stage-dependent transcription factor binding site (TFBS) prediction using genome-wide occupancy analysis has provided insights into mechanisms underlying the temporal activity of Satb2. Collectively, our data demonstrate that Satb2 functions in a versatile manner to influence crucial developmental transitions. Furthermore, its ability to participate in different biological contexts likely depends on diverse protein interactions and the dynamic repertoire of genomic targets.

## Results

**SATB family proteins diverged with the evolution of jawed fish.** To gain a detailed insight into the possible structural conservation among the SATB family proteins, we extracted the homologous sequence stretches by comparing their individual protein sequences. Initially, SATB family proteins were thought to be restricted to the vertebrate lineage[32]. Subsequently, a distantly related ortholog Defective proventriculus (Dve) was identified both in fruit flies and roundworms. Interestingly, our phylogenetic analysis revealed the presence of SATB homologs in two other invertebrates; Parasteatoda tepidariorum (spider) and Mizuhopecten yessoensis (a molluscan-scallop) which include all three functional domains (Fig. 1a and Supplementary Fig. 1a). While Agnatha, a class of vertebrate jawless fishes, exhibits a single form of SATB protein, the presence of both SATB1 and SATB2 is only apparent in jawed fish and later species. Consistently, previous reports have suggested possible function of SATB2 in the development of jaw, craniofacial structures and osteoblast differentiation in vertebrates[25,26,33]. Importantly, based on the domain architecture and phylogenetic affinities, SATB2 is closely related to the invertebrate homologs and hence can be regarded as the most ancient member of the family. Therefore, we focused on dissecting the function of SATB2 during the evolution of developmental transitions underlying body plan determination.

**Disruption of Satb2 function leads to defective craniofacial patterning.** In zebrafish embryos, compromising Satb2 activity using morpholinos resulted in severe developmental defects including epiboly arrest. These phenotypes were attributed to aberrant exocytosis and endocytosis[27]. We reasoned that the pleiotropic phenotype could be a result of depletion of the maternally deposited as well as zygotically transcribed Satb2. Thus, to decipher the maternal versus zygotic requirement of Satb2, we introduced a mutation in satb2 locus using the CRISPR-Cas9 system[34] by inserting a STOP cassette into the first exon, leading to premature termination (Fig. 1b and Supplementary Fig. 1c). The absence of Satb2 in the mutant fish was confirmed by immunoblot analysis (Fig. 1c, d).

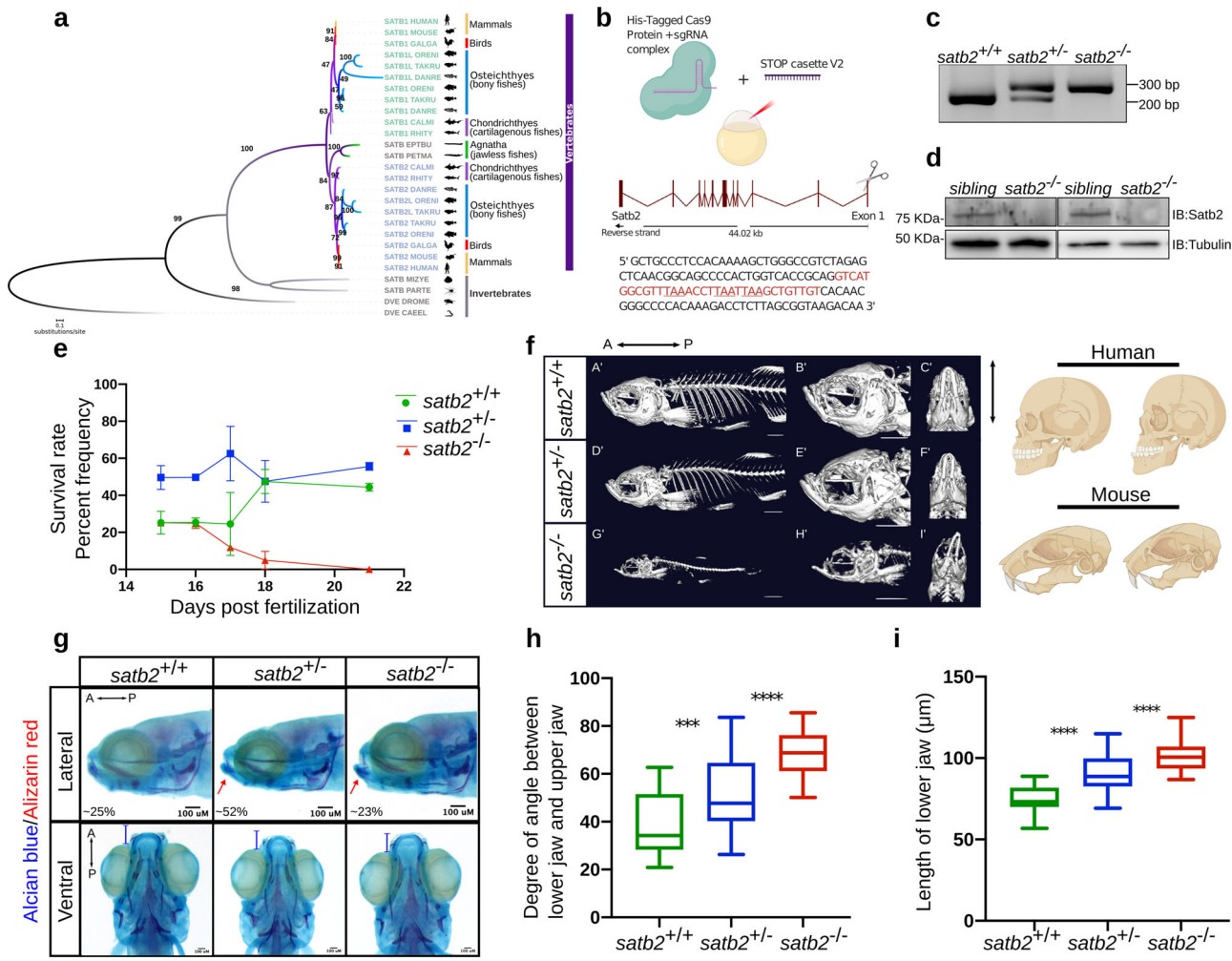

**Fig. 1 Loss of function of Satb2 in zebrafish leads to craniofacial defects and phenocopy function of mammalian homologues. a** Phylogenetic tree depicting evolution and divergence of SATB family proteins. SATB1 homologues are highlighted in green, SATB2 in blue whereas ancestors of SATB proteins are in gray. Organisms belonging to different classes are highlighted with different colors. Numbers on branching points represent bootstrap values. Organism labels: HUMAN- *Homo sapiens*, MOUSE- *Mus musculus*, GALGAL- *Gallus gallus*, ORENI- *Oreochromis niloticus*, TAKRU- *Takifugu rubripes*, DANRE- *Danio rerio*, CALMI- *Callorhinchus milii*, RHITY- *Rhincodon typus*, EPTBU- *Eptatretus burger*, PETMA- *Petromyzon marinus*, MIZYE- *Mizuhopecten yessoensis*, PARTE- *Parasteatoda tepidariorum*, DROME- *Drosophila melanogaster* and CAEEL- *Caenorhabditis elegans*. Additional copies of SATB1 and SATB2 were denoted with SATB1L (SATB1-like) and SATB2L (SATB2-like). **b** Schematic of CRISPR-Cas9 mediated mutant generation by introducing the STOP cassette (highlighted in red) in exon1 of the *satb2* gene resulting in loss of function mutant allele. **c** Identification of mutant allele by genotyping (60 bp insertion). $N = 5$ independent biological experiments. **d** Confirmation of loss of Satb2 between siblings and homozygous larvae at 48 hpf by immunoblot. Tubulin was used as a loading control. $N = 3$ independent biological experiments. **e** Lifespan analysis of zygotic *satb2* mutants depicted as a percentage frequency of individual genotype from a pool of larvae screened (incross of *satb2*$^{+/-}$) by a line graph. Wild-type (green), Heterozygous (blue), and Homozygous mutants (Red). Error bar represents ±S.D. of two independent biological experiments with $n = 48$ larvae. **f** Micro CT images of adult zebrafish (4-month old) to visualize defects in skeletal and craniofacial structures. Lateral view A′, D′, G′, magnified lateral view B′, E′, H′ and ventral view C′, F′, I′ of wild type, heterozygous and mutant for Satb2 respectively are shown. Ventral view images are not scaled to attain a maximum field of view highlighting detailed structural deformities independent of the size of the fish. Image is a representative of $N = 5$ individual organisms. Scale bar = 5 mm. Corresponding schematics on the right depicting similarity with *SATB2* mutation reported in humans and mice. **g** Alcian blue/Alizarin red staining to visualize craniofacial defects at early larval stages, 15 dpf. Abnormal jaw protrusion in heterozygous and homozygous mutants is indicated by a red arrow. Numbers in respective boxes signify the percentage of the larvae from the total population showing class of phenotype and genotype correlation. Scale bar = 100 μm. **h** Degrees of the angle between the lower jaw and upper jaw. *** indicates *p*-value = 0.002 and **** indicates *p*-value of significance 0.0001 as determined by the student's two-tailed *t*-test. The whiskers show the minima to the maxima values and the central line indicates the median, *satb2*$^{+/+}$, ($n = 24$) 38.04, *satb2*$^{+/-}$, ($n = 50$) 52.14, *satb2*$^{-/-}$, ($n = 22$) 68.57 over three independent biological experiments. **i** Total length of protrusion measured from ventral view as indicated in **g**, are plotted using a box plot. **** indicates *p*-value of significance 0.0001 as determined by the student's two-tailed *t*-test. The whiskers show the minima to the maxima values and the central line indicates the median, *satb2*$^{+/+}$, ($n = 26$) 74.68, *satb2*$^{+/-}$, ($n = 48$) 90.25, *satb2*$^{-/-}$, ($n = 22$) 102.0 over three independent biological experiments.

Homozygous zygotic *satb2* mutants showed poor survival to adulthood with most of the *satb2*$^{-/-}$ fish being depleted from the pool after 16 days post fertilization (Fig. 1e). However, we could occasionally recover homozygous survivors (~1%) from the total population ($n = 400$). Such survivors displayed severe deformities and reduced body size compared to their heterozygous siblings (Supplementary Fig. 1b, c). The homozygous fish were also sterile. Given the requirement of Satb2 during craniofacial development and bone formation in mammals, we recorded CT scans of homozygous survivors, heterozygous mutant siblings, and wild-

type fish ($n = 5$). *satb2* mutant fish displayed skeletal defects, deformed mandibles with a high degree of lower jaw protrusion, reduced maxilla with larger eye pockets, and abnormal skull size and shape (Fig. 1f). Supporting the haploinsufficient nature of Satb2 function, heterozygous fish also showed less severe yet detectable cranial structural abnormalities. Thus, zebrafish *satb2* mutant mimics the craniofacial defects observed in humans and mice[14] (Fig. 1f).

To trace back the onset of these defects during early development, we performed alcian blue: alizarin red staining on larvae obtained by mating *satb2* heterozygous adults. Larvae were collected 15 days post fertilization, a day prior to depletion of zygotic *satb2*$^{-/-}$ fish from the pool. As in the case of adult survivors, we observed craniofacial defects in the homozygous larval samples (Fig. 1g). We also detected an increase in the angle between the mandibles and maxilla in heterozygous and homozygous mutant larvae as compared to wild-type siblings (Fig. 1h). Measurement of the total length of the lower jaw confirmed significantly higher protrusion in zygotic *satb2* mutants (Fig. 1i). Our data demonstrate that zygotic loss of Satb2 in zebrafish leads to defective cranial and skeletal morphogenesis. Therefore, we decided to employ this unique tool to uncover its function during initial events of vertebrate organogenesis.

**Satb2 potentiates NC specification and skeletal system development.** Aberrant NC specification and migration have been correlated with severe craniofacial defects[29,30]. In the developing larvae, *satb2* is strongly expressed in the pharyngeal arches, a reservoir of NCCs[26]. This prompted us to investigate the possible involvement of Satb2 in NC development. First, we examined *satb2* expression in NCCs compared to non-neural crest cells from publicly available RNA-seq data[35] (Crestin$^{Pos}$ and Crestin$^{Neg}$ cells) (Supplementary Fig. 2a) and by qPCR on Sox10$^{Pos}$ and Sox10$^{Neg}$ cells isolated from Tg:Sox10:GFP line (Supplementary Fig. 2b) at the 14 ss. We observed very high enrichment of Satb2 transcripts in NCCs, suggesting the possible role of Satb2 in driving NC specification during very early stages of embryo development in zebrafish. To gain further insights, we performed gene expression analysis in wild-type siblings and *satb2*$^{-/-}$ embryos at three developmental time points that correspond to specific stages of NC differentiation (Supplementary Fig. 2c). To study NC induction, NC specification and differentiation we analyzed gastrulating embryos at 80% epiboly, embryos at 5–6 somites stage (ss), and 14 ss respectively[36]. Pearson correlation analysis for all stages showed significant reproducibility between the replicates (Supplementary Fig. 2d). Differential analysis revealed that loss of Satb2 function resulted in deregulation of numerous genes across all developmental stages (658 at 80% epi, 661 at 6 ss, 3015 at 14 ss) (Fig. 2a). Notably, we observed a minimal overlap between differentially expressed genes across developmental stages (Supplementary Fig. 2e). Furthermore, we observed maximum dysregulation of gene expression programs at 14 ss (both at the level of gene expression and in terms of the number of deregulated genes). This observation suggests that Satb2 likely plays an essential role during NC differentiation (Fig. 2a and Supplementary Fig. 2f). Supporting this conclusion further, at 14 ss, the significant deregulated targets were known components of NC development or differentiation such as *sox10*, *sox9b*, *pax3a*, and *zic1*[37–40] (Fig. 2b).

Consistent with the selective nature of Satb2's function, many paralogs within the gene families such as *prdm* (*prdm11*, *prdm14*), *foxp* (*foxp4*, *foxp1a*) and *snai* (*snai1* and *snai1b*) were differentially regulated by Satb2 (Fig. 2c). Gene ontology analysis of upregulated genes at 14 ss showed significant

enrichment in pathways involved in RNA polymerase activity, ribosome biogenesis and mRNA translation. Importantly, downregulated gene sets were enriched for the components of skeletal morphogenesis, NC differentiation and early embryonic developmental pathways (Fig. 2d). Supporting a conserved function of zebrafish Satb2, mutations in these target genes have been correlated with neurodevelopmental and craniofacial abnormalities in humans. To validate these changes, we performed whole-mount RNA in situ hybridization analysis for *id2a*, *sox9a* and *zic1*. *satb2*$^{-/-}$ mutants exhibited reduced expression of *sox9a* in the neural tube and *zic1* in the developing head. In contrast, *id2a*, a negative regulator of NCC specification, was considerably upregulated (Fig. 2e).

Satb2 has been implicated in cell type-specific activities during mouse neural development and has the potential to act both as an activator and a repressor[20,23,24,41]. To better understand the function of Satb2 during differential cell fate specification, we performed single-cell RNA-seq analysis on wild-type siblings (*satb2*$^{+/+}$) and mutants (*satb2*$^{-/-}$) at 14 ss (Supplementary Fig. 2g). UMAP clustering enabled the identification of 20 unique cell clusters (cluster0-cluster19) based on non-overlapping gene expression programs (Fig. 2f). Consistent with our phenotypic analysis and bulk RNA-seq dataset, upon loss of Satb2 function, we observed reduction in neural progenitor cell populations (cluster 0 and cluster 1) and cranial NC (cluster 8: marked by sox10 expression) (Fig. 2g). Moreover, although the number of lateral mesoderm progenitors showed an increase in the mutants, as in the case of bulk mRNA-seq data, *foxd3* expression in these cells was reduced while *snail1a* expression was upregulated (Fig. 2c). Altogether, these data underscore the regulatory potential of Satb2 to modulate differential gene expression triggering unique cellular identities. Our data also indicates that loss of Satb2 leads to reduction in endothelial precursors (*cdh5*$^{Pos}$) cells and increase in optic progenitors (*rx1*$^{Pos}$) cells (Supplementary Fig. 2h). Collectively, these findings show that zygotic Satb2 is necessary to regulate gene expression programs driving initial events of NC and neural progenitor cell specification.

**Genome-wide occupancy analysis confirms the direct regulation of downstream target genes by Satb2.** To investigate the molecular basis of Satb2 function, we performed chromatin immunoprecipitation followed by high throughput sequencing (ChIP-seq) for the corresponding stages of embryonic development. To this, we generated a polyclonal antibody against a peptide spanning the N-terminal region of Satb2 (see "Methods" and Supplementary Fig. 3a). To test the antibody, we overexpressed 3xFLAG-Satb2 from the one-cell stage and performed ChIP-seq at the dome stage using either anti-Flag or anti-Satb2 antibodies. Confirming the specific nature of the antibody, we observed a very high correlation (Pearson correlation 0.80) between ChIP signal intensities across these two conditions (Supplementary Fig. 3b). Next, we performed ChIP-sequencing in replicates using anti-Satb2 antibody. To identify Satb2 binding sites in open chromatin regions (OCR), we intersected consensus peaks with ensembl of chromatin accessibility data across various developmental stages [GSE106428, GSE130944, GSE101779] yielding 27,420 binding sites at 80% epiboly, 36,444 sites at 6 ss and 22,333 sites at 14 ss respectively. Genomic distribution analysis suggested that Satb2 occupies both the promoter and non-promoter regions (intergenic + intronic + exonic + TTS) (Fig. 3a). To gain insights into NCC-specific regulation by Satb2, we isolated Sox10$^{Pos}$ and Sox10$^{Neg}$ cells from Tg:Sox10:GFP transgenic line[42] using FACS and performed ChIP-seq for Satb2 at the 14 ss. Interestingly, we observed a strong correlation

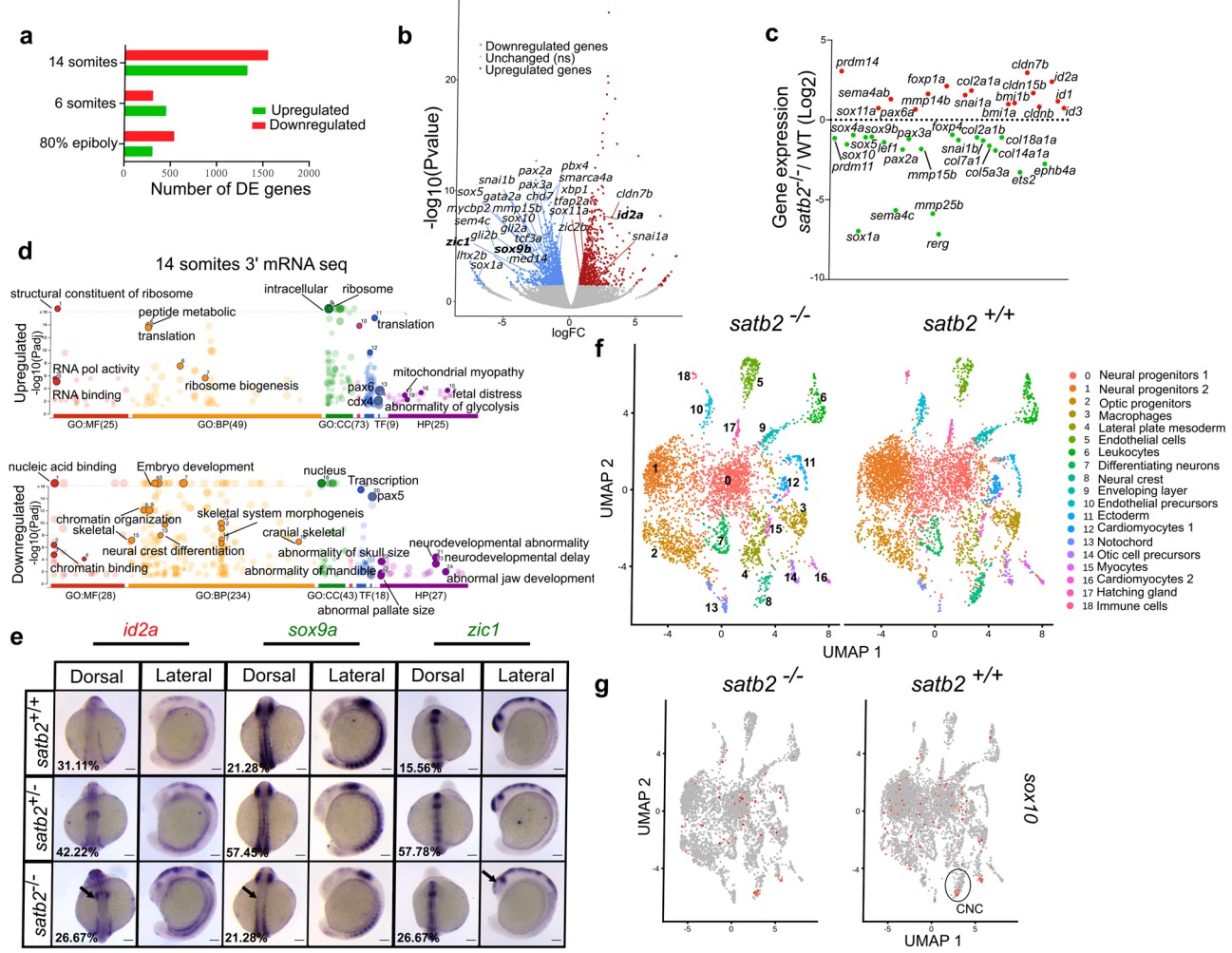

**Fig. 2 Deregulation of gene expression in *satb2* mutants at the onset of organogenesis leads to defective neurogenesis and craniofacial patterning.**
**a** Bar plot for the number of upregulated (red) and downregulated (green) genes from 3′ mRNA-seq datasets at the respective stages. **b** Volcano plot of differentially expressed genes at 14 ss embryonic stage. DE genes with FDR < 0.1 and log2 fold change > ±0.58 are colored as red for significantly upregulated, blue for significantly downregulated, and gray for unchanged or non-significant genes. Few significantly differentially expressed genes representing GO categories neural crest differentiation, neurogenesis, and cranial skeletal system development are highlighted. **c** Scatter plot for log2 fold change in gene expression of various gene families belonging to the above-mentioned GO categories. **d** GO analysis of dysregulated genes at 14 ss highlighting significant classes under molecular function (MF), biological processes (BP), cellular components (CC), transcription factors (TF) and human pathology (HP) categories. **e** Dorsal and lateral view of WISH for *id2a* (upregulated) n = 48, N = 3; *sox9a*, n = 47, N = 3 and *zic1*, n = 45, N = 3 (downregulated). Arrows in the respective images mark the region of maximum differences in the expression pattern. Scale bar = 100 μm. **f** UMAP clusters illustrate 19 distinct cell types through integrated scRNAseq analysis on *satb2⁻/⁻* and *satb2⁺/⁺* embryos at 14 ss. **g** UMAP highlighting (circled) cranial neural crest cells cluster expressing *sox10*.

between whole embryo Satb2 ChIP-seq at 14 ss and NC specific Satb2 ChIP-seq datasets (Supplementary Fig. 3c). This observation suggested that we can potentially utilize the bulk ChIP-seq approach to glean insights into gene regulation mediated by Satb2 specifically at the early stages of development when fluorescent reporter lines cannot be utilized. Indeed, the ChIP-seq occupancy profiles indicate that genes involved in early neurogenesis and NC development such as *sox10*, *sox9a*, *zic1*, and *zic4* are potentially book-marked for regulation by the presence of Satb2 as early as at 80% epiboly stage (Fig. 3b).

To assess whether Satb2 binds to putative enhancer regions of these genes, we intersected the Satb2 regulatory peak set with previously published dataset for poised enhancers across developmental stages [GSE32483, GSE74231] as marked by H3K4me1[43]. Satb2 peaks from the promoter and putative enhancer regions were further subjected to motif analysis. Interestingly, we observed a strong enrichment for transcription

factors including Pitx1, Tgif1, Foxk1, and Meis1 at the promoters (Fig. 3c). Mutations in these genes also affect craniofacial development in mice and humans[44–48]. Moreover, enhancer bound regions showed strong enrichment for binding sites of known regulators of NCC development such as Foxo1, Sox10, and Hoxa11[37,42,49,50]. To assess whether the binding of Satb2 influences gene expression, we compared corresponding transcript levels between wild-type and *satb2* mutants. Indeed, genes containing Satb2 binding sites within promoters and/or enhancers were significantly downregulated in the mutants (Fig. 3d).

We also performed reciprocal analysis comparing transcript levels between wild-type and mutants obtained from RNAseq experiments and correlated it with Satb2 occupancy. To achieve this, we annotated Satb2 bound regions to the nearest genes and analyzed the overlap between the two. This analysis demonstrated that Satb2 occupied both upregulated and downregulated sets of genes (Fig. 3e). Thus, we concluded that Satb2 exhibits bimodal

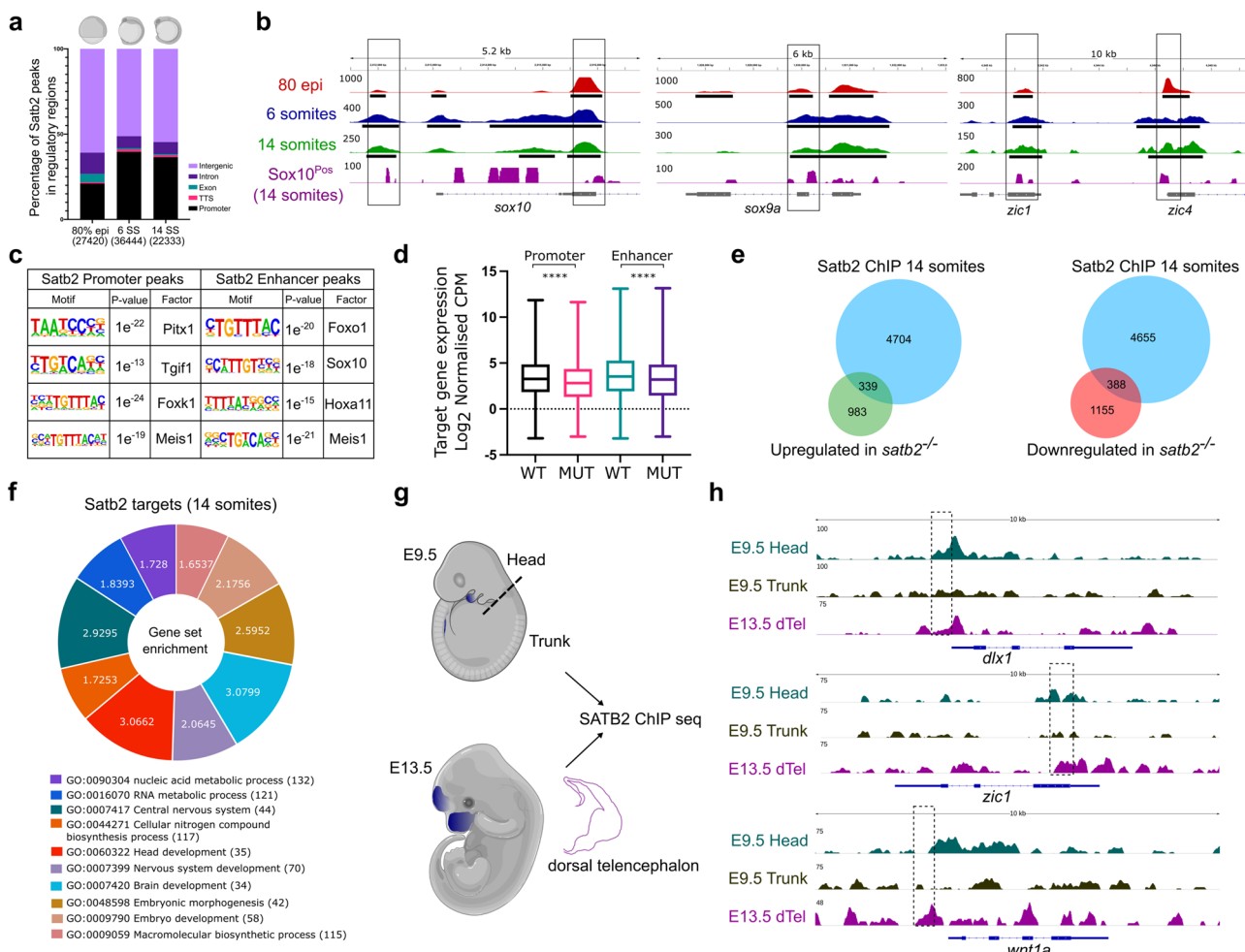

**Fig. 3 Genome-wide occupancy analysis of Satb2 underscores the molecular mechanism underlying Satb2 function. a** Genome-wide distribution pattern of Satb2 occupancy in the regulatory regions. percentage of peaks overlapping with OCR at 80% epiboly, 6 ss, and 14 ss. Numbers in the brackets indicate the total number of Satb2 peaks at the corresponding developmental stage. Annotation was performed using HOMER, core promoter region was defined as ±2 kb from TSS. TTS, Exon, and Intron annotations were used from HOMER analysis. The remaining peaks were annotated as Intergenic peaks. **b** Integrative genomics viewer (IGV) snapshot of Satb2 occupancy on genomic loci of *sox10*, *sox9a*, and *zic1* at 80 % epiboly (red), 6 ss (blue), 14 ss (green), and in Sox10[Pos] (purple) cells. Input subtracted tracks are used for visualization. For Satb2 ChIPseq in Sox10[Pos] (neural crest) cells, ChIPseq reads are normalized to signal from Sox10[Neg] (non-neural crest) cells to depict enrichment at respective gene loci. Solid line boxes highlight enriched genomic regions across all the datasets. Black colored blocks represent enriched regions determined by the peak calling algorithms. **c** Motif enrichment analysis for Satb2 binding sites at the promoter (±2 kb) and enhancer regions (overlapping with H3K4me1 binding sites) matched to known motif sets of danRer10 genome assembly. **d** Box plots indicating log2 normalized gene expression (Counts Per Million) of promoter- and enhancer- bound genes in wild-type and *satb2*[−/−] mutants at 14 ss. N = 2 biologically independent experiments. **** indicates *p*-value of significance 0.0001 as determined by the student's two-tailed *t*-test. Error bar represents ±SEM. The whiskers show the minima to the maxima values and the central line indicates the median, WT promoter (n = 4178): 3.347, MUT promoter (n = 4149): 2.890, WT enhancer (n = 3337): 3.662, MUT enhancer (n = 3327): 3.281. **e** Venn analysis depicting the overlap between Satb2 bound genes and differentially regulated genes at 14 somites in the s*atb2*[−/−] mutant. **f** GO analysis for biological processes of Satb2 occupied genes at 14 ss. Numbers for each category of doughnut plot represent enrichment ratio, FDR < 0.05. **g** Schematic of dissections performed for the ChIP-seq experiment in mouse embryos at E9.5 (head and trunk) and E13.5 (dorsal telencephalon). Dotted line represents the site of the incision. **h**, IGV snapshot of mouse SATB2 occupancy profile on the genomic loci of *dlx1*, *zic1*, and *wnt1*. Dashed line boxes highlight enriched regions in E9.5 head and E13.5 dTel which are absent in the E9.5 trunk region.

activity through directly binding to cis-regulatory sequences. To confirm this notion, we subjected Satb2 genomic targets to gene ontology analysis. Consistent with our earlier observation, the upregulated gene set showed enrichment for biological processes involved in nucleic acid metabolism, RNA metabolism, and nitrogen biosynthesis; whereas the downregulated gene set was enriched for processes involved in central nervous system development and morphogenesis (Fig. 3f). Taken together, the correlation analysis between Satb2 occupancy and mutant transcriptome underscores a distinct function of zygotic Satb2 in regulating early neurogenesis and RNA metabolism.

**Direct binding to cis-regulatory elements is a conserved function of Satb2.** To extend these observations, we decided to check if the expression pattern and function of Satb2 are conserved in higher mammalian systems such as mouse. We first analyzed previously generated RNA-seq datasets of cranial NCCs[51]. Indeed, *SATB2* is expressed as early as E8.5 in NC progenitor cells and its expression persists in NC subtypes such as frontonasal population (FNP), maxillary (Mx), and mandibular (Md) cells (Supplementary Fig. 3e). Moreover, single-cell analysis of cranial and trunk NCCs[52] revealed that *SATB2* is specifically enriched in cranial NCCs but not in the trunk NCCs

(Supplementary Fig. 3f). To examine if the restricted expression of *SATB2* is functionally relevant, we performed ChIP-seq analysis on developing mouse embryos. The use of developing mouse embryos provided us with an opportunity to assess the possibility of differential spatial regulatory control exerted by *SATB2* during NC specification. At E9.5 we dissected the head (source for cranial NCCs) and the trunk tissue (source of NCCs contributing to the peripheral nervous system). As a positive control for ChIP experiments, we isolated the dorsal telencephalon (dTel) at E13.5 to enrich the population expressing Satb2 in the developing cortex (Fig. 3g, Supplementary Fig. 3h). The ChIP-seq analysis in mice indeed showed significant enrichment for Satb2 on the regulatory regions of bonafide NCC markers such as *dlx1*, *zic1*, and *wnt1* specifically in the head tissue. However, Satb2 occupancy was not enhanced in trunk tissue for these regulatory regions, suggesting locus-specific and spatially restricted regulation by mouse SATB2 (Fig. 3h, Supplementary Fig. 3i). Taken together, our data suggest that functional regulation by Satb2 through direct binding at cis-regulatory elements of target genes is a conserved mechanism across vertebrates.

**Loss of Satb2 results in reduced chromatin accessibility and perturbed nucleosome positioning**. As Satb2 activity requires binding to target DNA sequences, we were interested in examining if Satb2 also modulates chromatin architecture. To assess this, we performed ATAC-seq analysis of wild-type siblings and $satb2^{-/-}$ mutants at 14 ss. $satb2^{-/-}$ mutants yielded a significantly lower number of OCRs. Interestingly, the percentage of OCRs at intergenic loci dropped significantly as compared to the promoters (Fig. 4a). Clustering (K-means = 4) of ATAC-seq signals around the ±2 Kb center of Satb2 peaks showed significant reduction across the genome in *satb2* mutants (Fig. 4b and Supplementary Fig. 4a). The gene ontology analysis revealed cluster 2 to be enriched for biological processes such as somite formation, embryonic morphogenesis, and craniofacial development (Fig. 4c) whereas cluster 1 showed enrichment for various metabolic processes (Supplementary Fig. 4b). Motif enrichment analysis of cluster 2 yielded significant overlap with binding sites for known regulators of neurogenesis and NCC specification such as Sox3, Pitx1, and Sox10 (Fig. 4d). To identify regions with differential chromatin accessibility, we subjected the ATAC-seq data to DiffBind analysis. We found a high number of regions (6861) with loss of chromatin accessibility in mutants as compared to only 141 regions with a gain of accessibility (Fig. 4e). Furthermore, we observed that modulation of chromatin accessibility primarily influences intergenic regions of genes including those involved in NCC specification (Fig. 4a, f, g).

Reduced chromatin accessibility is a critical determinant of gene regulation. However, genome organizer proteins can directly affect nucleosome positioning[53]. To test whether Satb2 has any impact on nucleosome positioning, we estimated nucleosome occupancy from ATAC-seq data using the nucleoATAC suit[54]. We observed reduced occupancy signals in mutants compared to wild type at the transcription start site (TSS) as well as at the Satb2 binding regions (Supplementary Fig. 4c, d). To analyze this further, we compared the nucleosome distribution at specific Satb2 target gene loci. We observed disruption in nucleosome phasing which resulted in fuzzy nucleosome patterns, particularly at the Satb2 binding sites (Fig. 4h and Supplementary Fig. 4e). Of note, we did not observe any such change in nucleosome positioning at the TSS for genes that are upregulated in Satb2$^{-/-}$ mutants (Supplementary Fig. 4c). These findings suggest that downregulation of Satb2 target genes could be resulting due to disruption in defined nucleosome phasing. In sum, the chromatin accessibility analysis uncovered multiple mechanisms through which Satb2 actively ensures sustained gene expression to drive neurogenesis and NCC development.

**Tracing back the activity of Satb2 during ZGA**. Data presented in the previous sections demonstrate a direct regulatory role of zygotic Satb2 during early neurogenesis and NCC development. Intriguingly, Satb2 is also maternally deposited and is thought to function primarily during epiboly[27]. Analysis of publicly available time-series transcriptome data during zebrafish embryogenesis[55] revealed bi-phasic expression of *satb2*. Maternal *satb2* is significantly reduced at the onset of ZGA and maintained at a minimal level before zygotic *satb2* starts expressing during organogenesis (Fig. 5a). To assess the function of maternally deposited *satb2*, we attempted to generate maternal-zygotic (MZ) mutants for *satb2*. As in the case of mice, zygotic *satb2* mutants show poor survival and the rare survivors are infertile[25] (Fig. 1e).

We next turned our attention to the biological relevance of the reduction of maternally deposited s*atb2* at the onset of a major wave of ZGA. To assess this, we overexpressed *satb2* from 1 cell stage (Supplementary Fig. 5a, b). Overexpression of Satb2 resulted in significant developmental defects as observed in the case of morpholino-mediated knockdown (Fig. 5b). This observation suggested that temporally controlled reduction of Satb2 could be biologically relevant. To gain further insight into Satb2 function, we carefully examined functional consequences of the sustained presence of Satb2 in early zebrafish embryos. Satb2 overexpressing embryos fail to survive beyond 8 hpf indicating possible early defects in cell-type specification and morphogenesis. Closer inspection of these embryos uncovered thickening of the dorsal organizer region. Similarly, in *Xenopus*, *satb2* expression has been observed to be enriched in the dorsal organizer suggesting its involvement in dorsoventral patterning[56].

Next, we performed transcriptome analysis of Satb2 overexpressing embryos. To rule out the effect of developmental delay on the gene expression we used embryos at 4.5 hpf (dome stage). As in the case of zygotic *satb2* mutants, overexpression of Satb2 also resulted in significant changes in gene expression (Supplementary Fig. 5c). However, in contrast to zygotic loss of Satb2, gene ontology analysis revealed that upregulated genes primarily constitute part of metabolic processes whereas downregulated genes were enriched in embryonic development, pattern specification, skeletogenesis, neurogenesis, and NC differentiation (Fig. 5c). These data argue that maternal Satb2 potentially acts as a repressor of the group of genes that are subsequently activated by zygotic Satb2 during organogenesis. (Fig. S5D).

To confirm this directly, we inactivated Satb2 function using two different morpholinos targeting the translation start site and splice junction resulting in early developmental arrest (Supplementary Fig. 6a). Five base mismatch morpholino was used as a control. Moreover, a morpholino-resistant version of *satb2* mRNA could partially rescue the phenotype (Supplementary Fig. 6b). Importantly, in contrast to Satb2 overexpression, knockdown embryos exhibit reduced expression of dorsal marker *chrd* (Supplementary Fig. 6c).

Next, we performed bulk mRNA-seq upon Satb2 knockdown and specifically analyzed the expression of genes involved in neurogenesis and NC development which are positively regulated by zygotic Satb2. As anticipated, markers for neurogenesis and NC development were prematurely upregulated (Fig. 5d and Supplementary Fig. 6d). However, compromising maternal Satb2 levels did not affect exocytosis and endocytosis pathway components[27]. These results establish that unlike zygotic Satb2, the maternal pool of Satb2 is involved in transcriptional repression and possibly regulates the same group of genes in an opposite manner.

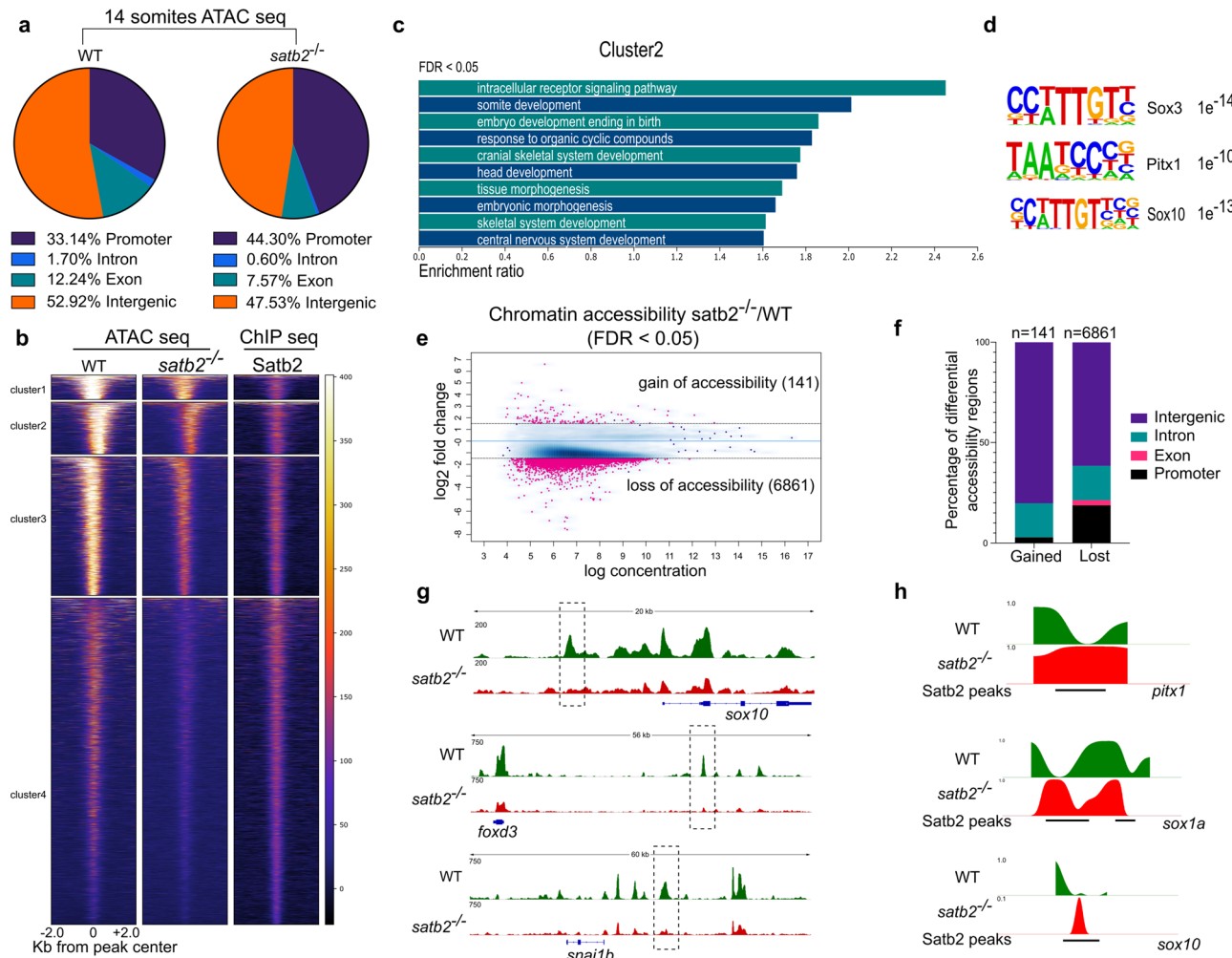

**Fig. 4 Loss of Satb2 affects chromatin accessibility. a** Genomic distribution of ATAC-seq peaks in wild type and *satb2*−/− mutant at 14 ss as a percentage of the total. Annotation was performed using HOMER, core promoter region was defined as ±2 kb from TSS. TTS, Exon, and Intron annotations were used from HOMER analysis. The remaining peaks were annotated as Intergenic peaks. **b** Heatmaps for clustering (K-means 4) of chromatin accessibility around ±2 kb of Satb2 ChIP-seq peak regions. Colorbar represents the degree of chromatin accessibility from low (blue) to high (white). **c** GO analysis of cluster 2 highlighting biological processes involved in neural and craniofacial development. **d** Enriched TFBS are represented for cluster 2. **e** Differential chromatin accessibility between wild-type and *satb2*−/− mutant embryos represented as binding affinity (FDR < 0.05). Y-axis represents log2 fold change. Regions with log2 fold (>±0.58) were considered significant and marked by solid lines. **f** Genome-wide distribution of differentially gained or lost regions in *satb2*−/− mutant embryos represented as a percentage. **g** IGV snapshot of ATAC-seq peaks over the genomic loci for enhancer regions of neural crest markers *sox10*, *foxd3* and *snai1b*. Dashed line boxes highlight regions that are identified as differentially expressed through Diffbind analysis. **h** Nucleosome occupancy profile calculated using ATAC-seq showing changes in nucleosome phasing at Satb2 regulatory sites for *pitx1*, *sox1a* and *sox10*.

**Satb2 functions in a cell-autonomous manner**. To assess the nature of Satb2's function during ZGA we employed single-cell gene expression analysis. We overexpressed Satb2 in a mosaic manner by injecting 3xFLAG-Satb2 in one of the cells at 16 cell stage embryos and harvested embryos at 4.5 hpf to prepare single-cell suspension (Fig. 5e). This strategy also enabled us to obtain wild-type cells from the same source. We generated a high-quality dataset for 3′ gene expression in 8483 single cells using scRNA-seq (Supplementary Fig. 5e, f). UMAP clustering of these cells identified 7 distinct clusters (cluster0 to cluster6) based on the correlation of gene expression patterns (Fig. 5e). Next, we analyzed the expression of individual genes as specific features and generated spatial maps for *satb2* and developmental genes which are found to be negatively regulated by Satb2 through bulk mRNA-seq analysis. To exemplify, we have represented the spatial maps for *satb2*, *ta*, *mixl1*, *krt8*, and *aplnrb* and highlight a negative correlation between the expression of *Satb2* and these developmental genes (Fig. 5f). In conclusion, mosaic

overexpression followed by scRNAseq indicates that Satb2 functions in a cell-autonomous manner.

**Maternal Satb2 modulates local chromatin landscape**. Lee et al. have previously characterized early developmental genes into strictly maternal, strictly zygotic, and first-wave zygotic genes[8]. Upon overexpression of Satb2, genes belonging to strictly zygotic, and first wave zygotic genes were significantly downregulated (Fig. 5g and Supplementary Fig. 5g). By contrast, maternally deposited genes required to support early growth were upregulated[57] (Fig. 5c, g). Next, to assess if repression of zygotic genes is correlated with corresponding epigenetic changes, we profiled the occupancy of H3K4me3 (activation mark), H3K27me3 (repressive mark), H3K27Ac (activation mark) and analyzed chromatin accessibility (ATAC-seq) focusing on TSS of 'zygotic strict' genes. Curiously, we observed depletion and increase in the activation and repressive marks respectively despite a very minimal effect on chromatin accessibility (Fig. 5h).

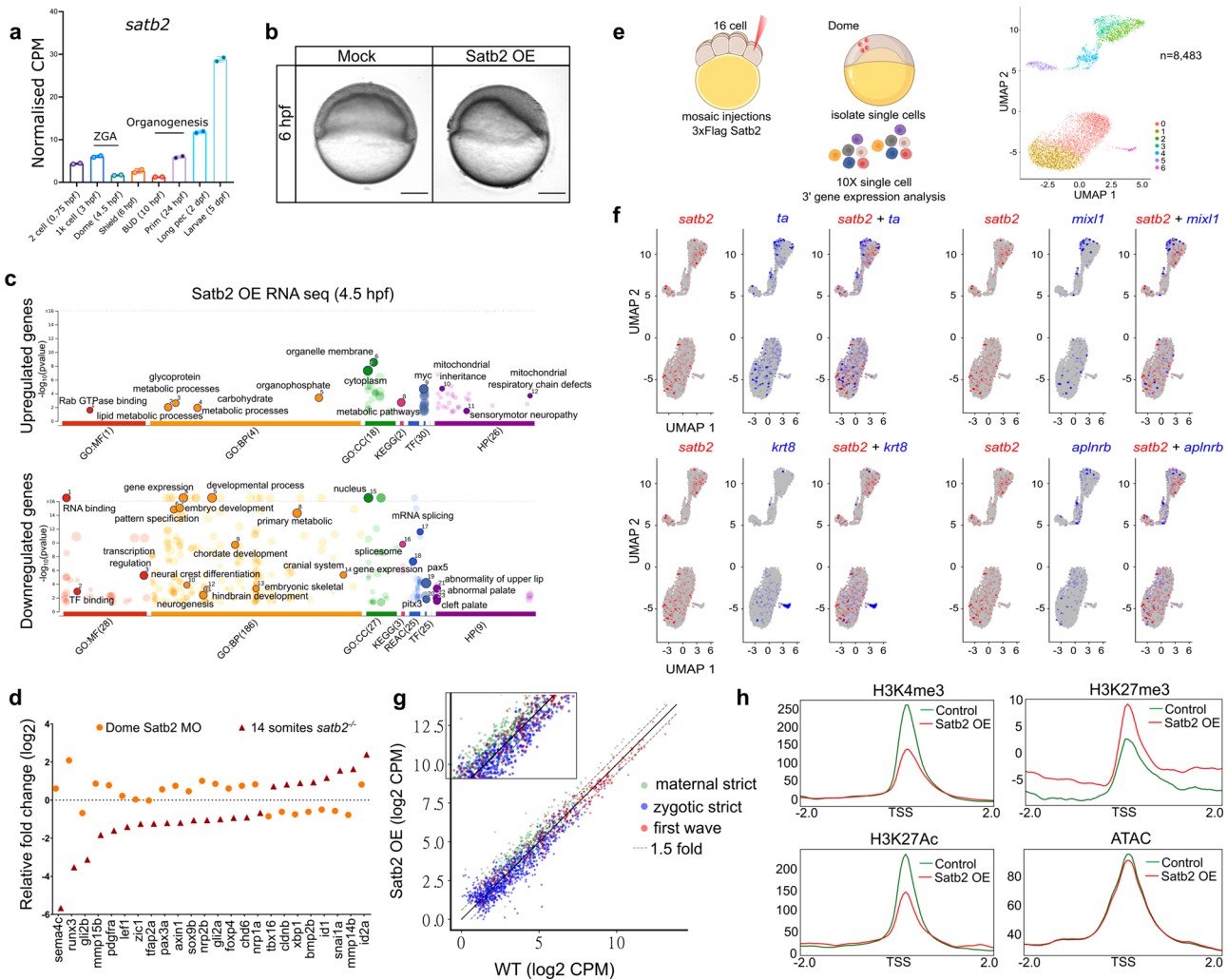

**Fig. 5 Maternal Satb2 functions as a transcriptional repressor prior to ZGA. a** Stage-specific gene expression analysis highlighting bi-phasic expression of Satb2 during MZT and organogenesis. Normalized CPM values for each stage are used for the analysis [GSE32900]. N = 2 biologically independent experiments. Error bar represents ±SD. **b** Lateral view of zebrafish embryos at 6 hpf injected with 200 pg eGFP (Mock) and 3xFlag-satb2 mRNA (Satb2 OE). Scale bar = 100 μm. N = 4 biologically independent experiments. **c** GO analysis for differentially expressed genes upon over-expression of Satb2 at 4.5 hpf. Significant ontologies under molecular function (MF), biological processes (BP), cellular components (CC), reactome (REAC), transcription factors (TF), and human pathology (HP) categories are highlighted. Numbers in brackets signify the total number of ontologies enriched under each category. **d** Relative expression values for genes involved in neural crest differentiation upon morpholino mediated depletion of maternal Satb2 (yellow circles) analyzed at 4.5 hpf and in zygotic satb2$^{-/-}$ mutants (red triangles) analyzed at 14 ss, highlighting opposite effects on gene regulation by maternal and zygotic forms of Satb2 (FDR < 0.1). **e** Experimental strategy for single-cell analysis of embryos with mosaic overexpression of Satb2 at 4.5 hpf. UMAP clustering illustrates eight different clusters based on gene expression patterns from 8483 single cells. **f** FeaturePlots for exemplifying negative correlation between cells expressing satb2 (blue) and zygotic factors ta, mixl1, krt8, and aplnrb (red). **g** Biplot depicting the effect of overexpression of Satb2 on maternal (green), zygotic (blue) and first wave zygotic genes (red). log2 CPM values are used for the analysis. The dotted line marks genes that show more than 1.5-fold differences in log2 CPM values. Inset is a zoomed version to highlight the differences. **h** Average mean density profiles around ±2 kb region around TSS of 'zygotic genes' for H3K4me3 (activation mark), H3K27me3 (repressive mark), H3K27Ac (activation mark) and ATAC-seq (chromatin accessibility) upon Satb2 overexpression (red) compared to control (green).

This could be presumably due to the transient regulation of the epigenetic landscape or active compensatory mechanisms.

**Interplay between Satb2 and pioneer factors at ZGA.** Role of the pioneer factors such as Pou5f3 (homolog of OCT4), Nanog, and SoxB1 has been well established in activating ZGA[8,9]. To test if maternal Satb2 is preventing precautious transcription of zygotic patterning genes by targeting pioneer factor(s), we investigated if these factors are specifically regulated by Satb2. Supporting the notion, RNAseq analysis, and RNA whole-mount in situ hybridisation demonstrated downregulation of pou5f3. Notably, unlike pou5f3, sox19b was upregulated upon

overexpression of Satb2, suggesting that the influence of Satb2 on the pioneer factors is qualitatively non-uniform (Fig. 6a, b). Interestingly, many of the direct targets of Pou5f3 and Sox2 showed significant overlap with negative targets of Satb2 (Fig. 6c, d). We subsequently confirmed that the negative regulation between Satb2 and Pou5f3 is not unique to zebrafish but is conserved across vertebrates (Fig. 6e and Supplementary Fig. 7a). To verify if the negative regulation is mediated by direct binding of Satb2, we performed ChIP-seq at a pre-MBT stage (512 cells) and post MBT stage (dome) (Supplementary Fig. 7b). We observed strong enrichment of Satb2 at the TSS of both pou5f3 and sox19b (Fig. 6f). However, ChIP-qPCR analysis

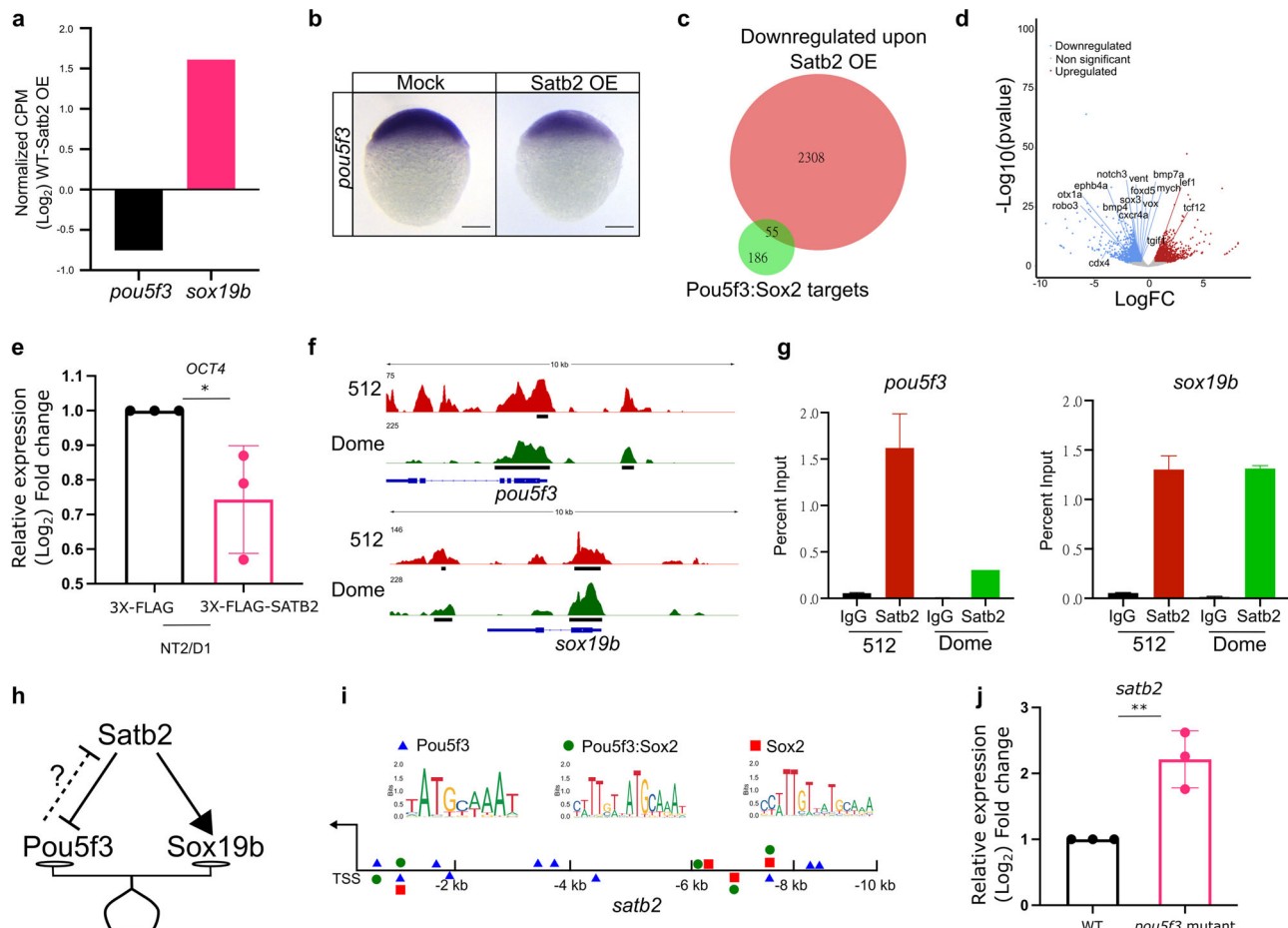

**Fig. 6 Maternal Satb2 functions during ZGA by regulating pioneer factors. a** Relative gene expression for *pou5f3* and *sox19b* as determined by RNA-seq analysis (FDR < 0.05). **b** Lateral view of zebrafish embryo at 4 hpf for WISH using RNA probe against *pou5f3* in control and Satb2 overexpressing embryos, n = 24, N = 3 biologically independent experiments Scale bar = 100 µm. **c** Venn diagram analysis showing the overlap between negative targets of Satb2 and known genomic targets of Pou5f3 and Sox2. **d** Volcano plot of differentially expressed genes upon overexpression of Satb2 analyzed at 4.5 hpf. DE genes with FDR < 0.1 and log2 fold change > ±0.58 are colored as red for upregulated, blue for downregulated, and gray for unchanged or non-significant genes. Key Pou5f3: Sox2 target genes are labeled. **e** Quantitative RT-PCR analysis of *OCT4* (human homologue of *pou5f3*) expression in NT2/D1 cells upon overexpression of SATB2 as compared to FLAG overexpression. * Signifies P-value 0.049 as calculated by the unpaired Student's two-tailed t-test, N = 3 biologically independent experiments. Error bar represents ±SEM. **f** IGV snapshot of Satb2 ChIP-seq at 512 cells (red) and Dome stages (green) on genomic loci of *pou5f3* and *sox19b*. **g** Quantitative relative enrichment (ChIP-qPCR) represented as percent input using isotype-matched IgG and anti-Satb2 antibody for genomic locus (TSS) of *pou5f3* and *sox19b*. ** indicates P-value < 0.001 as calculated by the Student's two-tailed t-test, N = 2. **h** The predictive regulatory model between maternal Satb2 and pioneer factors during the establishment of ZGA. **i** Schematic for *satb2* promoter displaying motif sites marked for Pouf53, Pou5f3: Sox2 and Sox2. **j** qPCR analysis for *satb2* expression at 4.5 hpf in *pou5f3* mutant embryos. ** Signifies P-value < 0.0082 as calculated by the unpaired Student's two-tailed t-test, N = 3 biologically independent experiments. Error bar represents ±SEM.

showed reduced Satb2 occupancy on the genomic *pou5f3* locus from pre- to post- MBT but not on the promoter of *sox19b* (Fig. 6g). Such dynamic occupancy pattern was also observed for other negative targets of Satb2 including *ta*, *bmp4*, and *gsc* (Supplementary Fig. S7c). Altogether, these results support that depletion of maternal *satb2* is essential for sustained expression of the pioneer factor *pou5f3* and in turn the onset of ZGA. However, we were curious as to how the maternal pool of *satb2* is regulated at the onset of ZGA. Since Pou5f3, Nanog and SoxB1 act as pioneer factors, we asked if there is a feedback regulation between these factors and Satb2 (Fig. 6h). To test this, we first scanned the *satb2* promoter (up to −10 Kb) for binding motifs of Pou5f3: Sox2 complex, Pou5f3 alone, and Sox2 alone. A number of potential binding sites were observed for these pioneer factors within a 2 Kb window upstream to TSS (Fig. 6i). Previously published ChIP-seq data at post-MBT stages supported this observation (Fig. S7E). To establish the functional relevance of

the binding of Pou5f3, we performed expression analysis of *satb2* using qRT-PCR in *pou5f3* mutants. Significant upregulation of *satb2* thus supports the presence of possible negative feedback regulation between Satb2 and the pioneer factor Pou5f3 (Fig. 6j).

**Early developmental regimes established using human embryonic stem-like cells recapitulate biphasic nature of *SATB2* expression.** Temporal analysis of Satb2 expression revealed an interesting biphasic pattern that translated into bimodal regulation. We wondered if this characteristic pattern can be mimicked using pluripotent NT2/D1 embryonic cells. Upon retinoic acid (RA) stimulation, NT2/D1 cells differentiate into neuronal progenitors[58]. We found that while *SATB2* is readily detectable in undifferentiated cells, *SATB2* levels decrease during transition stages (day 3 post RA treatment). Interestingly, we also observed an increase in *SATB2* expression as cells

differentiated into neuronal progenitors, characterized by an increase in *PAX6* expression (Supplementary Fig. 8a). Next, we analyzed the transcriptome of H9 human ES cells differentiation series[59]. Even in this scenario, *SATB2* is initially transcribed in the pluripotent cells, and gets downregulated upon initiation of differentiation from day 2 to day 6. *SATB2* transcription is restored during later stages (day 8 and beyond) when neuronal fate specification ensues (Supplementary Fig. 8b).

**Stage-specific occupancy analysis of Satb2 reveals the underlying mechanisms for the differential activity of maternal and zygotic Satb2.** Our results highlight the contrasting nature of maternal versus zygotic Satb2 function during early embryogenesis. Maternal Satb2 primarily functions as a repressor while zygotic Satb2 activates neurogenesis programs. Such disparate behavior could be attributed to two potential mechanisms: (1) Differential occupancy at the target loci and/or (2) Stage-specific distinct interacting protein partners. Towards this, we evaluated Satb2 occupancy by revisiting our ChIPseq datasets from 512 cells stage to 14 ss. Interestingly, we observed a substantial shift in the ratio of non-promoter to promoter-bound regions by Satb2 (Fig. 7a). Maternal Satb2 occupies more of the non-promoter regions (67% intergenic and 10% promoter) at the pre-MBT stage, whereas zygotic Satb2 localizes substantially to the promoter regions (54% intergenic and 36% promoter) of the same target genes.

Next, we performed consensus binding analysis for non-promoter and promoter-bound regions at 512 cell stage and 14 ss to identify putative interactors of Satb2. Maternal Satb2 occupied regions were enriched for binding sites of Ebf1, Ear2 (nr2f6), Hnf4a, Coup-TFII (Nr2f2), Ap2-gamma (Tfap2c) and Erra. Among these, Ebf1, Ear2, Coup-TFII proteins have been widely characterized for repressive activity during B cell and NC differentiation[60,61] (Fig. 7b). In contrast, zygotic Satb2 binding regions show enrichment for Hoxa11, Bmyb, Tgif1, Scl, Sox10, and Meis1, all essential factors for NC differentiation and neurogenesis[42,45,50,62–64].

Expression dynamics of these transcription factors highlight putative stage-dependent association with Satb2 (Fig. 7c, d). Using the STRING database and published literature, we extracted putative interactors of these proteins. Potential interactors of maternal Satb2 display propensity to be associated with co-repressors such as Hdac1, Myc, and Med1[65–67] whereas interactors of zygotic Satb2 are likely to partner with the positive regulators of neurogenesis including Pitx1, Tcf3a, Dlx3a, Tgif1, Olig2, and Zic2a[45,68–72]. This analysis argues that the contrasting stage-dependent function of Satb2 is a consequence of qualitatively distinct, unique sets of binding partners (Fig. 7e). However, future experiments will be necessary to test this idea.

## Discussion

Early embryonic development is a culmination of carefully calibrated gene expression patterns orchestrated in a spatiotemporal manner. Typically, this outcome is achieved by establishing tissue-specific GRNs. How the GRNs are initiated, maintained, and executed has been a major focus of inquiry among developmental biologists. Consequently, we have learned a great deal about the regulation of tissue-specific gene expression leading to cell fate specification. In recent years the focus has gradually shifted to the elucidation of genome-wide molecular changes that engineer critical cellular/developmental landmarks such as ZGA or early determination of cell-type identity. However, how the global changes in genome architecture are mechanistically connected to individual loci has remained mysterious. Equally unknown are the individual players that participate as crucial

molecular links between these processes. Here, we have investigated the function of the chromatin organizer protein Satb2 in developmental transitions during early vertebrate embryogenesis.

Studying Satb2 in this regard is especially pertinent as it appears to be the most ancient member of its cohort across evolutionary scale and its function is required during embryogenesis for proper specification of NCCs, a cell type critical for craniofacial development[18,25,73]. Indeed, the zebrafish *satb2* mutants generated in this study recapitulates the phenotypes seen in the corresponding aberrations in mice and also resembles the human genetic conditions induced by the disruption of *SATB2* locus. Emboldened by this similarity we embarked on a detailed analysis of *satb2* mutant embryos using functional genomics. We simultaneously employed analogous samples using mouse tissues and human cell lines and, thus far, several observations of broad significance have emerged.

Based on deep RNA sequencing and scRNAseq analysis, *SATB2* was found to be enriched in mouse NC progenitor cells[51,52]. We also observed similar enriched expression of *satb2* in zebrafish cranial NCCs. Based on in-situ hybridization analysis, Ahn et al. suggested that *satb2* is widely expressed during the early stages of embryogenesis[27]. Our quantitative RNA-seq approach refined these observations and provided extensive information about spatial expression patterns of *satb2*. Interestingly, Satb2 binds to multiple cis-regulatory sequences in a genome-wide manner. Moreover, Satb2 can act both as a repressor and as an activator of transcription.

Through our comprehensive transcriptome, genome-wide occupancy, and chromatin accessibility studies, we have identified several positive and negatively regulated targets of Satb2. scRNA seq analysis of Satb2 mutants further allowed us to gain important insights into cell type-specific effects of loss of function of Satb2. For instance, Satb2 activates transcription of the effectors of neural progenitors and NC specification while it negatively regulates the components of ribosomal biosynthesis and endothelial and lateral mesoderm progenitors. Thus, it would be of interest to investigate the functional relationships between the putative target genes of Satb2 to elucidate the circuitry underlying NCC specification. In fact, such cross regulatory interactions that fine-tune gene expression via feed-forward and feed-back mechanisms is an established hallmark of embryonic development.

Through this study we highlight the functional dichotomy between the maternally deposited and zygotically synthesized Satb2. Maternally deposited Satb2 diminishes at the onset of ZGA before resuming function during zygotic development including organogenesis. Intrigued by this biphasic nature of *satb2* expression, we analyzed the function of maternal Satb2 by employing morpholino-based knockdown and overexpression. This biphasic nature of Satb2 was likely missed in the earlier studies as these approaches were based on less quantitative methods[27]. However, in depth analysis of temporal RNA-seq datasets allowed us to discern the unique expression pattern and its functional dynamics. Our analysis of the pluripotent embryonic cells uncovered that maternal Satb2 acts as a transcriptional repressor of zygotic genes involved in patterning and likely contributes to the prevention of the premature onset of ZGA. Moreover, under these circumstances, metabolic pathways appeared to be upregulated which could be either a direct or an indirect consequence of perturbation in ZGA.

Interestingly, loss of Satb2 in the murine system does not lead to early embryonic defects[74]. Such differences can be potentially attributed to species-specific gene functions or compensatory mechanisms as documented in the case of pluripotency factor OCT4. For instance, zebrafish homolog Pou5f3 and human OCT4 have been shown to regulate ZGA in zebrafish and

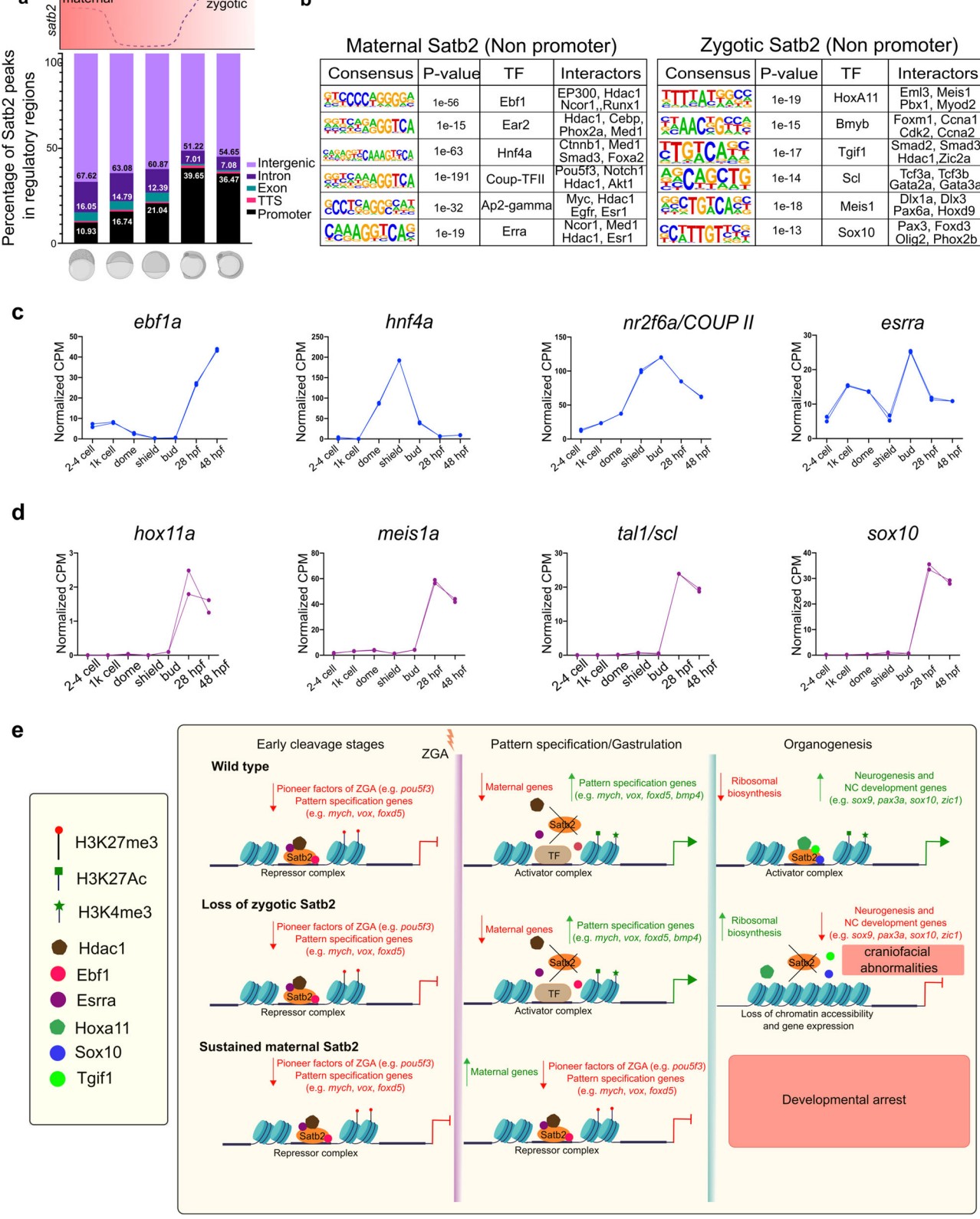

**Fig. 7 Differential mechanisms employed by maternal and zygotic Satb2 to perform contrasting functions during embryogenesis. a** Genomic distribution of Satb2 bound regions across developmental stages highlighting a shift in the ratio of non-promoter to promoter bound peaks. Schematic above the bar plot represents the expression pattern of *satb2* throughout early embryogenesis. **b** TFBS for transcription factors at maternal (512 cells stage) and zygotic (14 ss) Satb2. Known interactors for each transcription factor are enlisted. **c** Line graph depicting mRNA expression dynamics for putative interactors of maternal Satb2; *ebf1a*, *hnf4a*, *nr2f6a*, and *esrra* across developmental stages as indicated on the x-axis. **d** Line graph depicting mRNA expression dynamics for putative interactors of zygotic Satb2 namely, *hox11a*, *meis1a*, *tal1*, and *sox10*, across developmental stages as indicated on the x-axis. **e** Schematic summary highlighting differential functions of Satb2 during various stages of zebrafish embryogenesis.

humans respectively. However, OCT4 does not regulate ZGA in mouse[8,9,75]. Moreover, residual NANOG activity could potentially compensate for the loss of Satb2 in mouse embryos[74]. It is also noteworthy that these studies have not systematically categorized the effect of loss or gain of function of SATB2 on maternally deposited versus zygotically synthesized gene targets. Previous studies have also alluded to a possible antagonism between the activities of SATB1 and SATB2[17,74,76]. Interestingly, such antagonism was not observed in our analysis during zebrafish embryogenesis. Thus, through this study, we characterized multiple important functions of Satb2, few of which are unique to zebrafish, while others are largely conserved across the vertebrates.

Supporting the idea, unlike the zebrafish specific function of Satb2 during early ZGA phase, the role of zygotically synthesized Satb2 seems to be highly conserved. The biphasic expression of Satb2 is instrumental in manifesting bimodal, contrasting molecular functions. Perturbation of ZGA by Satb2 overexpression resulted in severe developmental defects during zebrafish embryogenesis. Consistent with our observations at 14 ss, we found that maternal Satb2 also regulates the local chromatin landscape of target genes by modulating histone modification occupancy profiles. Reprogramming of histone modification patterns is essential for successful onset of ZGA[77,78]. These studies indicate that maternally deposited H3K27me3 is depleted prior to the onset of ZGA with concomitant increase in H3K4me3 and H3K27Ac. Interestingly, we did not observe significant perturbation in chromatin accessibility. Thus, we propose that during cleavage stages, Satb2 brings about transient chromatin modulations rather than generating lasting effects through active chromatin remodeling.

The bimodal activity of Satb2 poses two major questions. What regulates the expression of Satb2 during embryonic development? And how does Satb2 acquire differential regulatory potential? BMP and SHH signaling regulate the expression of *satb2*[18,28]. Since these signaling pathways are dormant during early cleavage stages and thus the upstream regulators of maternal Satb2 need further investigation. We also report a negative feedback regulation between Satb2 and pioneer pluripotency factors such as Pou5f3. Importantly, we observe critical target selection during this process as Satb2 negatively regulates Pou5f3 but positively regulates Sox19b to maintain gene expression homeostasis necessary for ZGA. Such a mechanistic interplay has been proposed by Gao et al., providing insights into how the balance between Pou5f3 and Sox19b is essential for efficient ZGA[69]. Interestingly, we do not observe regulation of Nanog by Satb2 as previously documented during mouse embryonic development[17]. Again, this distinction may likely be due to species-specific differences in mode of gene regulation and differential target selection. Subsequent comparative studies will allow us to gain further insights into the evolution of these processes. Through our study, we propose that Satb2 potentially acts as a determinant of organism-specific GRNs by regulating one or more of the "Yamanaka" factors during the process of ZGA.

Intriguingly, we report that maternal Satb2 acts as a transcriptional repressor of the same group of genes which are later activated during organogenesis. Thus, we sought to analyze underlying mechanisms for contrasting functions of maternally deposited and zygotically synthesized Satb2. Stage-dependent ChIP-seq analysis revealed a dynamic shift from the non-promoter- to promoter-bound fraction of Satb2 on the same target genes. These observations along with motif analysis suggested that Satb2 potentially regulates target genes in a context-dependent manner based on its genomic occupancy and the potential to form a complex with multiple partners (Fig. 7c, d). Such mechanisms have been proposed for the bimodal activity of

NC fate determinant Foxd3[79]. Moreover, the gene regulator function of SATB1 is known to be governed in a bimodal fashion via interaction partners in T cells[80].

In summary, we have characterized temporally regulated context-dependent functions of Satb2 (Fig. 7e). Our studies have unraveled the involvement of Satb2 as a gatekeeper during major regulatory events throughout early vertebrate embryogenesis. This study also underscores Satb2 as a bonafide member of a group of proteins that wear multiple hats and function in a similar biphasic and bimodal manner during body plan determination.

## Methods

**Experimental models**. Zebrafish (Danio *rerio*) were maintained as described in the zebrafish handbook[81]. All the experimental procedures were carried out in accordance with the guidelines from the institutional animal ethics committee at IISER Pune and IST Austria. Embryos were raised at 23–31 °C in E3 medium and visually staged[82]. Experiments with mice models (*Mus musculus*) were carried out at TIFR, Mumbai adhering to the Institutional ethics guidelines.

**Phylogenetic analysis of SATB proteins**. Homologous sequences for SATB proteins were extracted from GeneTree data from Ensembl [http://www.ensembl.org/Multi/GeneTree/Image?gt=ENSGT00390000008096] and were further curated using NCBI BLASTP in selected organisms for their true copy in their genome to avoid inclusion of splice variants and genomic copy numbers. SATB1 splice variants were removed from the analysis, however, no splice variants were observed for SATB2. Curated sequences were aligned using MUSCLE using default parameters and alignments were trimmed using trimAl on auto mode which are embedded in the online server [https://ngphylogeny.fr][83]. A phylogenetic tree was generated using the randomized accelerated maximum likelihood method (RAxML) method available at CIPRE Science Gateway server[84]. Here, PROTCAT was used as a substitution model with DAYHOFF substitution matrix. DVE CAEL was used as an outgroup and 1000 bootstraps were performed for the analysis. The resulting phylogenetic tree was visualized using iTOL and tree branches are color coded based on different groups[85]. The representative animal's silhouette images were collected from [http://phylopic.org/] and used in annotation.

**CRISPR/Cas9 mutant generation**. Synthetic oligonucleotides targeting exon1 of *satb2* gene using CRISPR-Cas9 method were designed using CHOPCHOP and cross-validated using CRISPRscan[86,87]. Oligonucleotides required to generate sgDNA synthesized commercially (Supplementary Table 2) (IDT, USA).

sgDNA template was generated using gene-specific oligonucleotides and constant oligonucleotides by polymerase chain reaction for 30 cycles using Q5 high fidelity polymerase (NEB, USA). The PCR product was further purified using MagBio Hiprep PCR cleanup system (MagBio Genomics, USA) and quantified using Nanodrop 2000. sgRNA was synthesized in vitro as per the suggested protocol using Ampliscribe T7 Flash Transcription kit (Lucigen, USA). The 6xHis-Cas9 protein was synthesized in vitro[33]. Briefly, plasmid coding for 6x-His-Cas9 were transformed into Rosetta DE3 Novagen competent cells (Merck Millipore, USA). Protein production was induced using an autoinduction medium (Formedium, UK) for 36 h at 18 °C with constant shaking at 200 RPM. Bacterial cells were harvested and lysed in 20 mM Tris pH 8.0, 30 mM Imidazole, 500 mM NaCl followed by sonication with 20 s ON 20 secs OFF pulses for 30 min using a probe-based sonicator, Sonics Vibra cell (Sonics, USA). Lysates were cleared using high-speed centrifugation and incubated with HisPur cobalt resin (Thermo Scientific, USA) for 1 h. Beads were further washed three times with lysis buffer and eluted in 20 mM Tris pH 8, 200 mM Imidazole, 500 mM NaCl. Eluted fractions were further dialyzed in 20 mM Tris, 200 mM KCl, 10 mM MgCl$_2$ and stored as 5 ul aliquots at −80 °C for further use.

The sgRNA-Cas9 protein complex (sgRNA 200 ng/μl, Cas9 600 ng/μl and STOP cassette 1 μM) was injected at the one cell stage of zebrafish embryos. Founders were obtained by genotyping using oligonucleotides flanking the target site and maintained as heterozygous.

**Genotyping of Satb2 mutants**. The embryos used for all the experiments were a result of satb2$^{+/-}$ incrosses. As a result, genotyping of the embryos after every experiment was necessary. DNA from embryos/larvae at desired stages of development were extracted in 50 μl of genomic DNA extraction buffer (10 mM Tris-HCl pH 8.0, 50 mM KCl, 0.3% Tween-20 and 0.3% NP-40) containing 10 μg proteinase K at 55 °C for 16 hrs. Proteinase K was inactivated by incubating at 95 °C for 20 min followed by snap chilling. For adult fish, genotyping was performed using fin clips[34]. PCR was carried out using paq polymerase 5000 (Agilent, USA), and resulting PCR products were electrophoresed on 3% agarose gel. Gel images were captured using G:Box gel documentation system (Syngene, USA).

**Micro-CT image acquisition and analysis.** $satb2^{+/-}$ fish were incrossed and were grown until 4 months post fertilization. Fish were fin clipped and genotyped using the protocol described above. Screened fish ($N = 5$ per group) were fixed in a 3.7% paraformaldehyde (PFA) and 1% glutaraldehyde in 1× PBS (137 mM NaCl, 2 mM KCl, 8 mM $Na_2HPO_4$, 1.5 mM $KH_2PO_4$) solution for 72 h at room temperature (RT) and then transferred to 1% of the fixative solution for storage at 4 °C. Fish were imaged first using SLR cameras and then were imaged using Quantum GX microCT imaging system (Perkin Elmer, USA). Image acquisition settings were as follows: Temperature: 23 °C, Humidity: 57%, Voltage: 90 kVp, Current: 88 μA, X-Ray dosage: 449 mGy, FoV: 36 mm, Recon: 36 mm, Voxel size: 72 μm, Scan mode: high resolution, Time: 4 min. Images were further processed with ImageJ (FIJI) for adjusting brightness and contrast.

**Alcian blue - Alizarin red double staining.** Alcian blue: Alizarin red acid-free double staining was performed to visualize bones and cartilages and bones respectively[88]. All solutions were stored, and all steps were carried out at RT. Briefly, embryos from satb2$^{+/-}$ incrosses were grown to 15 dpf and fixed with 4% PFA for 2 h. After dehydration for 10 min in 50% ethanol, larvae were stained overnight in a double staining solution made from a mixture of Alcian blue: Alizarin red stock solution in a 100:1 ratio respectively. Alcian blue stock solution for cartilage staining contained 0.1% Alcian blue powder (Sigma Aldrich) in 70% ethanol and 60 mM $MgCl_2$ solution. Alizarin red (Sigma Aldrich) stock solution for bone staining was made at a final concentration of 0.5% in MilliQ water. After staining, the solution was removed by a quick rinse with water followed by washes with 1:1 solution of 3% $H_2O_2$ and 2% KOH for 20 min with the tube caps kept open. Larvae were then cleared with successive washes with increasing concentration of glycerol (20–50%) and 0.25% KOH solutions. Larvae were finally stored in a 50% glycerol and 0.1% KOH storage solution until imaging. At the time of imaging, larvae were indexed and temporarily mounted with the help of 75% glycerol and 0.1% KOH solution and imaged using an Olympus light microscope. After three subsequent washes with PBST to remove the bone stain, selected larvae were genotyped for representation.

**3′mRNA gene expression assay and differential gene expression analysis.** Single embryos were harvested in 100 μl RNA iso-plus total RNA extraction reagent (DSS Takara, India) at 80% epiboly, 6 ss and 14 ss stages of zebrafish embryonic development. Samples were homogenized by vigorous vortexing to assure complete lysis followed by the addition of 15 μl of $CHCl_3$. Aqueous layer was separated by centrifugation at 12,000 × g for 15 min at 4 °C. The aqueous layer was collected in a separate tube and an equal volume of isopropanol containing GlycoBlue coprecipitant (Invitrogen, USA) was added. The aqueous fraction was stored at −20 °C until further processing. Meanwhile, the organic fraction was processed for DNA extraction by precipitating with 150 μl of 100% ethanol and incubated for 10 min at RT. Samples were centrifuged for 10 min at high speed and the pellet was washed with 70% ethanol (twice). After drying, DNA pellets were resuspended in low TE and used for genotyping as described before. Once the genotype for each embryo was confirmed, the corresponding RNA fractions were processed further. RNA from 2 to 3 samples were pooled to get enough input to prepare libraries.

Extracted RNA samples were quantified using Nanodrop 2000 and RNA integrity was determined using Agilent bioanalyzer 2100. Samples with RNA integrity number (RIN) > 8 were used to prepare sequencing libraries using Quantseq 3′ mRNA seq library kit (Lexogen GMBH, Austria) as per the instructions from the manufacturer. Briefly, 300 ng of RNA was used for poly-A capturing and subjected to first and second-strand synthesis. All cDNA samples were amplified for 10–15 cycles depending on cycle number estimation by qPCR (PCR add-on kit, Lexogen GMBH, Austria). Amplified libraries were purified using two rounds of 0.8x volume of Hi-prep PCR purification kit (MagBio Genomics, USA). The concentration of libraries was estimated using the Qubit dsDNA HS system (Thermo Fisher Scientific). Finally, all the libraries were pooled and subjected to high throughput sequencing using 76 bp SE chemistry on Nextseq 550.

Sequencing reads were trimmed for quality using Trimmomatic and aligned to the daRer10 genome assembly using STAR aligner. Counts for each gene feature were estimated using the FeatureCounts package from Rsubread. Differential expression analysis was performed for replicates using EdgeR. Volcano plot for significantly differentially expressed genes was generated using the PlotVolcano tool from Galaxy server. Gene ontology analysis for upregulated and downregulated genes was performed using a web-based tool, gProfiler[89].

**Synthesis of riboprobes for whole mount in situ hybridization.** Synthetic oligonucleotides against target mRNAs (Supplementary Table 2) with T7 RNA polymerase promoter sequence on the 5′ end of the reverse primers were obtained commercially (Sigma Aldrich, India). cDNA templates for corresponding stages were used for PCR amplification. PCR reactions were purified using NEB Monarch DNA Gel Extraction Kit or HighPrep PCR Clean-up System beads. Further, Riboprobes were synthesized using Roche DIG RNA labeling kit at 37 °C for 3 h followed by DNAse treatment for 15 min. Probes were further purified using Bio-Spin 30 Tris Columns (Biorad, Germany). The digoxigenin-labeled probes were 400-700 nucleotides in length and stored at −20 °C at 1 μg/ml.

For *chrd* WISH experiments, the amplified region was cloned into TOPO II dual promoter vector (Thermo Scientific, USA). To generate an antisense probe, plasmid was linearized using BamHI and probe synthesis reaction was carried out using T7 RNA polymerase promoter as described above.

**Whole mount in situ hybridization analysis.** Whole-mount in situ hybridization (WISH) was performed to study the localization of mRNA[90]. Fertilized embryos from $satb2^{+/-}$ incrosses were raised until the desired stage of development was reached. Embryos were fixed in 4% PFA in 1× PBS overnight at 4 °C with gentle rocking. Next day, embryos were hand dechorionated with the pair of fine forceps and incubated in 100% methanol overnight at −20 °C. Embryos were further downgraded in methanol: PBST solution (0.1% Tween-20 in 1× PBS pH 7.4). Following successive rehydration to PBST, embryos were treated with 5 μg/μl proteinase K solution in PBST for 2 min for 14 ss stage embryos. The reaction was stopped by a subsequent 20 min incubation in 4% PFA. For the dome stage, embryos were not treated with proteinase K. After washes in PBST, samples were pre hybridized for 2–5 h and incubated overnight at 70 °C in ~500 ng of the respective probe. Next day, embryos were washed with SSC (composition) buffers and further blocked using blocking buffer (2% v/v normal goat serum, 2 mg/ml BSA in PBST) for 3-4 h at RT followed by overnight incubation at 4 °C in anti-DIG-AP Fab fragments (1:5000) (Roche, #1093274). Embryos were washed with PBST followed by an alkaline tris buffer. Staining was performed with BM Purple substrate (Roche). After staining, the color reaction was stopped using a stop solution containing 0.1% Tween-20 and 1 mM EDTA in 1× PBS (pH 5.5). Embryos were further washed with PBST and stored at 4 °C until imaging. Embryos for imaging were mounted on agarose molds and imaged using Z-stacking mode on a Leica DFC450C microscope. Post imaging, embryos were genotyped using the standard protocol discussed above. Representative images were further processed using ImageJ for adjusting brightness and contrast.

**Single-cell RNA seq analysis of Satb2 mutants.** To understand the cell type-specific effects upon loss of Satb2 function, we analyzed the wild type siblings and $satb2^{-/-}$ embryos by single-cell RNA sequencing (scRNA-seq). Briefly, embryos from the cross of $satb2^{+/-}$ fish were raised to 14 ss at 28.5 °C, hand dechorionated and dissociated further in 1× DMEM-F12 (Sigma Aldrich) + 5 mM EGTA by hand tapping. Cells were collected by centrifugation at 300 × g for 2 min and resuspended in 1× DPBS (Gibco) + 0.1% BSA (Himedia). Genotype of each embryo was assessed by genomic PCR as described above. Cell suspension was passed through 70-μm Flowmi cell strainers (Sigma). Cell viability was estimated using CountessII automated cell counter and samples with more than 95% viability were processed further with 10X Genomics Chromium. Approximately 8000 cells were captured in GEMs. scRNA-seq libraries were prepared as described in 10X Genomics manuals (Single Cell 3′ Reagent Kits V3.2). Immediately following GEM generation, reverse transcription reaction was carried out using eppendorf mastercycler pro by incubating at 53 °C for 45 min followed by denaturation at 85 °C for 5 min. Single-stranded cDNA was purified using DynaBeads MyOne Silane Beads (Thermo Fisher Scientific, USA). cDNA amplification was performed for 11 cycles with initial denaturation at 98 °C 3 min followed by repeated cycles of 98 °C for 15 s, 63 °C for 20 s and 72 °C for 1 min. Final extension was performed at 72 °C for 1 min. cDNA quality was determined using Bioanalyzer 2100 dsDNA high sensitivity assay kit (Agilent). cDNA was further fragmented, end repaired and A-tailed according to the manufacturer's instructions. Before proceeding for adapter ligation, samples were purified using double-sided clean up protocol with SPRI beads (Beckman Coulter). Adapter ligated sample was subjected to amplification for 11 cycles using indexing primer from Chromium i7 Multiplex Kit (PN-120262). Amplified libraries were again subjected to double-side purification using SPRI and quantified using Qubit fluorometer (Thermo Fisher Scientific, USA) and the library size was estimated using Bioanalyzer 2100.

1.5 pM of the denatured libraries were used as an input to obtain sequencing reads using 28 cycles for read1, 8 cycles for indexes, and 101 cycles for read2 on Nextseq 550 (Illumina) at IISER Pune.

Sequencing data were further processed with default parameters using cell Ranger 6.0 (10X Genomics). Sequencing reads were aligned to the danRer10 genome using STAR aligner. Feature matrices generated using cell Ranger were utilized for further integrated analysis using Seurat 4.0.

**Antibody generation.** All the procedures were performed as per the approved guidelines from the ethical committee at the National Toxicology Centre (NTC), Pune. To generate polyclonal antibodies, anti-zebrafish Satb2 (c-term): CIPSS-GAEENPQANTGSGNNGP and anti-human SATB2 (c-term): CQQSQPA-KESSPPREEAP peptides were synthesized commercially (Apeptide, China). Antibodies were produced in New Zealand white Rabbits, as per the protocols from the laboratory of Tony Hyman, MPI-CBG with modification as follows [https://hymanlab.mpi-cbg.de/hyman_lab/general/]. Briefly, the required quantity of peptides was conjugated with KLH (keyhole limpet hemocyanin) using glutaraldehyde followed by subsequent dialysis to remove glutaraldehyde. Conjugated peptides were mixed with Freud's complete adjuvant (Sigma Aldrich) for the first immunization. Rabbits were immunized intradermally. Further, after every 21 days,

rabbits were immunized using peptides mixed with Freud's incomplete adjuvant until sufficient titer for the antibody was obtained.

Antisera was purified by a peptide affinity column prepared using sulfoLink coupling resin according to the manufacturer's instructions (Thermo Fisher Scientific, USA) and stored in 50% glycerol solution at −20 °C.

**Molecular cloning of expression plasmids**. For zebrafish Satb2 over-expression studies, full-length *satb2* was amplified from cDNA samples generated using High-capacity cDNA synthesis kit (Thermo Fisher Scientific, USA) from 48 hpf wild-type TU embryos. Resulting PCR product was purified using Monarch PCR purification kit (NEB, USA) and was cloned into pCS2+ vector using restriction enzymes BamH1 and SnaB1 to generate pCS2-Satb2. To generate 3xFLAG-tagged construct, an oligo containing 3xFLAG sequence at the 5′ end of the forward primer was used for PCR using pCS2-Satb2 plasmid as a template and clone was obtained using Gibson assembly. To generate morpholino resistant clones, oligo containing 7 mismatches spanning the first 24 bases of the coding region were used for amplification using pCS2-Satb2 plasmid as a template.

For recombinant expression of zebrafish Satb2 protein, coding sequences were amplified using Q5 high fidelity DNA polymerase (NEB, USA) and cloned into 6xHis-pET28-Strep vector using Gibson assembly. The PCR product was treated with DpnI and transformed into *E. coli* DH5α bacterial strain for in-vivo ligation. Resulting clones were screened by colony PCR. Recombinant protein was expressed in autoinduction medium (Formedium, UK) for 36 h at 18 °C with constant shaking at 200 RPM.

To obtain human SATB2 expression construct, coding sequence for human SATB2 was amplified from the IMAGE cDNA clone MGC:119475 IMAGE:40007830 using gene-specific primers and was cloned into EcoRI and XbaI digested p3XFLAG-CMV-9 vector (Sigma Aldrich, USA). Sequences of all the expression constructs were confirmed using Sanger sequencing and validated for expression using immunoblotting experiments. For each experiment, an empty vector was used as control as described in figure legends. Synthetic oligonucleotides used for cloning were obtained from Sigma or Eurofins and are listed below.

**mRNA and morpholino injections**. In-vitro mRNA transcription was performed using SP6 mMessage mMachine Kit as per the instructions from the manufacturer (Ambion). Synthesized mRNA was treated with Turbo-DNAse and precipitated using LiCl overnight at −20 °C. Synthesized mRNA was checked for quality and quantity by agarose gel electrophoresis and Nanodrop2000 respectively. Small single-use aliquots of mRNAs were kept frozen at −20 °C till further use. Glass capillaries (WPI) were pulled using a needle puller (P-97, Sutter Instruments) and mounted on a microinjection system (PV820, World Precision Instruments).

For ubiquitous overexpression and morpholino-mediated knockdown studies, embryos were arranged in agarose molds and injections were performed at 1 cell stage using agarose molds[81]. For overexpression studies, 200 pg of either pCS2-Satb2 or pCS2-3XFLAG-Satb2 were injected at 1 cell stage. For single-cell gene expression analysis, injections were performed with 200 pg of 3xFLAG-Satb2 at 16 cell stage targeting one cell per embryo to generate mosaicism. For rescue experiments, 5 pg of morpholino resistant Satb2 version was injected at 1 cell stage followed by injections of 4 ng morpholinos.

Custom Morpholino sequences (Supplementary Table 2) targeting translation initiation (MO1) and targeting exon splicing (MO2) were synthesized from Genetools and 1:1 mixture was used for all knockdown experiments. Five base mismatch morpholino was used as a control for all the experiments. Phenotypic changes were captured at indicated time points using Olympus stereo microscope.

**Genome-wide occupancy analysis for zebrafish Satb2**

*ChIP setup*. To map the genome-wide occupancy of zebrafish Satb2, ChIP sequencing was performed at the desired stages of embryogenesis as mentioned in the results section and figure legends. To validate the use of in-house raised antibody for successful ChIP-seq experiment, we injected embryos with 25 pg of 3xFLAG-Satb2 at 1 cell stage and harvested embryos at 4.5 hpf for further processing. ChIP was performed using anti-FLAG antibody (Sigma) and anti-zebrafish Satb2 antibody and subjected to high throughput sequencing as described below. Correlation matrix was generated to confirm the reproducibility of the two methods. For all stage-specific ChIP experiments, anti-zebrafish Satb2 was used to capture endogenous genomic binding regions.

Briefly, ChIP was performed using modified protocol[9]. Approximately, 2000 embryos per stage were harvested in 1× E3 medium (5 mM NaCl, 0.17 mM KCl, 0.33 mM CaCl₂, 0.33 mM MgSO₄, 0.1% methylene blue) and homogenized using glass homogenizer loose piston in the presence of 1 mM PMSF and immediately fixed with 1% methanol free formaldehyde (Thermo Fisher Scientific, USA) at RT for 12 min with constant shaking. Fixation was stopped by the addition of glycine to a final concentration of 0.125 M and incubation at RT for an additional 5 min. Fixed cells were centrifuged at 500 × *g* for 5 min and washed thrice with ice-chilled 1× PBS by centrifugation at 4 °C for 5 min each. Cells were further lysed in a six times bed volume in the cell lysis buffer (10 mM Tris-HCl pH 7.5, 10 mM NaCl, 0.5% (v/v) NP-40 for 10 min on ice and subjected to homogenization in Dounce homogenizer for 10 times with loose piston followed by three times with tight piston. Nuclei were collected by centrifugation at 2500 × *g*, washed with ice-cold

PBS and resuspended in eight times the pellet volume in the nuclei lysis buffer (50 mM Tris-HCl pH 7.5, 10 mM EDTA, 1% (w/v) SDS, 1× protease inhibitor cocktail (PIC: Roche,11873580001)) and incubated further for 30 min on ice. The samples were sonicated using the following Covaris S2 sonication conditions: 20% duty cycle, intensity = 5, cycle per burst = 200, time = 40 cycles of 30 s ON 30 s OFF to obtain an average size of 200–300 base pairs. The samples were centrifuged at 15,000 × *g* for 10 min and the chromatin containing supernatant was stored at −80 °C till further use. Prior to ChIP setup, supernatant was precleared using a mixture of 1:1 Dynabeads protein A:G beads for 2 h at 4 °C. For each ChIP replicate and mock reaction, 100 µg of chromatin was diluted 1:10 with ChIP dilution buffer (16.7 mM Tris-HCl pH 7.5, 167 mM NaCl, 1.2 mM EDTA, 0.01% (w/v) SDS, 1x PIC) and incubated with either 5 µg of anti-FLAG antibody or 20 µg of anti-Satb2 antibody. For mock ChIP reactions, an equal amount of Rabbit IgG (31235, Invitrogen) was used. Chromatin-antibody complex was incubated overnight at 4 °C with constant rotation. Following day, the Chromatin-Antibody complex was captured using 100 µl pre-blocked (with IgG-free BSA and t-RNA) Dynabeads® Protein A:G mix for 4 h at 4 °C. Beads were washed with ice-cold buffers in the following order: 7 min four times with low salt buffer (20 mM Tris HCl pH 8.0, 150 mM NaCl, 2 mM EDTA, 0.1% SDS, 1% Triton X-100), 10 min twice with high salt buffer (20 mM Tris HCl pH 8.0, 200 mM NaCl, 2 mM EDTA, 0.1% SDS, 1% Triton X-100), 10 min once with LiCl buffer (0.25 M LiCl, 1 mM EDTA, 10 mM Tris HCl pH 8.0, 1% NP-40, 1% sodium deoxycholate) and 10 min twice with TE buffer (10 mM Tris HCl pH 8.0, 1 mM EDTA). The TE buffer was removed completely and 150 µL of the elution buffer (0.1 M NaHCO₃, 1% SDS) was added and vortexed gently to solubilize the beads, followed by incubation at 65 °C for 30 min at 1000 rpm. The eluate was collected in a fresh tube and the process was repeated with the addition of 150 µL of the elution buffer. To 300 µL of the eluates and 10% input, 20 µL of 5 M NaCl and 2 µL of RNAseA (10 mg/ml) was added and the samples were incubated at 65 °C overnight with constant shaking at 700-800 rpm. Next, 20 µL of 1 M Tris pH 8.0, 10 µL of 0.5 M EDTA, and 2 µL of Proteinase K (20 mg/ml) was added and the samples were further incubated at 42 °C for 1 h at 700–800 rpm. Samples were purified by standard phenol:chloroform extraction method and DNA was precipitated overnight with equal volume of 100% isopropanol in the presence of Glycoblue (Ambion) at −20 °C. DNA pellets were eluted in nuclease-free water and quantified using Qubit fluorometer (Thermo Fisher Scientific, USA) before proceeding further.

To perform NC specific ChIP-seq for Satb2, Sox10[Pos] and Sox10[Neg] cells were isolated from Tg:Sox10:GFP transgenic fish using high-speed flow cytometry sorting (FACS ARIA III, Becton Dickinson). Cells were directly lysed in the lysis buffer to isolate chromatin by sonication. ChIP experiment was performed as described in the above section.

*Library preparation, sequencing, and data analysis*. An equal amount of DNA (~5 ng) was used as input for library preparation using Nugen ultra2 ovation kit (Nugen). The number of cycles for amplification of adapter-ligated libraries was estimated by the qPCR method as described in the datasheet provided by the manufacturer. Final libraries were purified using HiPrep PCR clean up system (MagBio Genomics, USA). Library concentration was determined using Qubit and average fragment size was estimated using dsDNA HS assay on Bioanalyzer 2100 (Agilent) before pooling libraries at equimolar ratio. Sequencing reads (100 bp PE) were obtained using the HiseqX platform at Macrogen Inc, Korea or Nextseq-550 at IISER Pune.

Sequencing reads were trimmed using TrimmomaticPE for Truseq2:PE adapters and read with quality greater than phred 33 were retained[91]. The quality of sequencing reads were determined using fastQC. High quality sequencing reads were aligned to the zebrafish danRer10 genome version using default parameters of BWA[92]. Aligned reads were subsampled to 40 million reads in each sample using BBMap[93]. Correlation between each replicate was estimated using multiBamSummary and replicates showing very high Pearson correlation (>0.7) were used for further analysis. Peak calling was performed using MACS2 with default parameters and q value 0.05. Consensus sets of peaks from biological replicates were extracted using a custom R script from Roman Cheplyaka [https://ro-che.info/articles/2018-07-11-chip-seq-consensus]. BigWig files were generated first using bamCoverage normalizing to RPKM and then subtracting Input signals using bamCompare utilities from deepTools 3.3.2[94] and used for visualization with Integrative Genomics Viewer (IGV). Satb2 peaks in regulatory genes were extracted by intersecting with Ensembl of ATAC seq peaks [GSE106428, GSE130944, GSE101779]. Further, Satb2 peaks were classified as putative enhancer bound peaks by intersecting Satb2 peaks with a consolidated dataset of H3K4me1 [GSE32483, GSE74231]. Promoter bound peaks were assigned as ±2 Kb from the TSS of the genes. Peak annotation to the nearest gene was performed using annotate.pl utility from Homer. Motif discovery for a given set of peaks was performed using findMotifsGenome.pl script from Homer and gene ontology was performed using webGestalt utilizing KEGG and Panther databases.

**ChIP qPCR**. ChIP experiment was performed as described above at the desired stages in replicates as indicated in figure legends. An equal amount of IPed DNA was used as an input for quantitative real-time PCR analysis. qPCR was performed using primers (Supplementary Table 2) with KAPA SYBR green master mix (Kapa biosystems) on ViiA7 Real-time PCR system (Thermo). Relative enrichment was

calculated using percent input method using the formula $100 \times 2^{\wedge}$ (Adjusted input − Ct (IP)).

## Genome-wide occupancy analysis for mouse SATB2

*ChIP setup.* To obtain genome-wide binding sites for SATB2 during early neurogenesis, SWR/J mice were incrossed and embryos were isolated at desired stage of development. For experiments with E9.5 head tissue and trunk (visceral organs were discarded) were dissected under stereo microscopes with fine forceps and collected in ice-chilled 1× PBS containing 0.5% glucose. Tissues from 10 to 12 embryos were pooled together for each replicate and crosslinked using 1% methanol free formaldehyde for 8 min followed by quenching with 125 mM glycine for an additional 5 min at RT. For experiments at E13.5, dorsal telencephalon was isolated and processed as described above. Chromatin isolation and ChIP experiment was performed as described in experiments for zebrafish Satb2 with modifications as follows. Hundred mucrograms of precleared chromatin was used for each IP reaction and 10 μg of anti-Human SATB2 antibody was used for pulldown. An equal amount of anti-Rabbit IgG (Invitrogen) was used as a control for each ChIP experiment. Antibody-protein complex was captured using 100 μl of pre-blocked Dynabeads protein A:G mixture for 4 h at 4 °C. Samples were processed further as described in the earlier section.

*Library preparation, sequencing, and data analysis.* Equal amount of DNA (~5 ng) was used as an input for library preparation and libraries were prepared using NEB ultra II DNA library prep kit (NEB, USA). Sequencing reads (76 bp PE) were obtained on the Nextseq 550 platform at IISER, Pune.

Sequencing reads were trimmed using TrimmomaticPE for Truseq2:PE adapters and were aligned to mouse mm10 genome using default parameters of BWA. Aligned reads were subsampled to 25 million reads for each sample using BBMap. QC, peak calling, and BigWig generation was performed as described in the above sections.

## ChIP Western

Efficiency of ChIP for mouse SATB2 and validation of mammalian SATB2 antibody was performed by ChIP western assay. Briefly, ChIP was performed with in house anti-human Satb2 antibodies. After bead washing and removal of excess TE buffer, beads were resuspended in 1× PBS and treated with 2 μl of DNAse (10 mg/ml) for 30 min at 37 °C. Further, beads were washed with 1× PBS twice and boiled in 2× Laemmli buffer (0.25 M Tris-Cl pH 6.8, 1% SDS, 1% β-mercaptoethanol, 15% glycerol) at 98 °C for 5 min and electrophoresed on 7.5% SDS-PAGE gel. Proteins were transferred to a 0.45 μM PVDF membrane (Millipore, USA) by wet transfer method at 0.6 A for 2.5 h at 4 °C. Non-specific sites on the membrane were blocked using 5% BSA and further incubated with anti-SATB2 antibody (1:1000 abcam ab34735) in 0.3% BSA overnight at 4 °C with constant rocking. Next day, the membrane was washed with 1× TBST (50 mM Tris-Cl pH 7.4, 150 mM NaCl, 0.1% Tween-20) four times for 7 min each and incubated with anti-rabbit HRP conjugated antibody (1:10,000 STAR124P, Bio Rad, USA) in 1X TBST for 45 min at RT. After removing excess secondary antibodies by repeated washes with 1× TBST, signal was developed with Clarity western ECL substrate (Biorad) and captured using the LAS 4000 system (GE Healthcare, USA).

## ChIP sequencing for histone modification marks

*ChIP setup.* To profile the status of histone modifications upon overexpression of 3xFLAG tagged Satb2, 1000 embryos were harvested for each ChIP reaction at 4.5 hpf (Dome stage) in batches until sufficient amount of input material was generated. Empty pCS2-GFP was used as a control for injections. Pulldowns were performed with 20 μg chromatin and 5 μg anti-H3K4me3 (ab8580, abcam), 5 μg anti-H3K27Ac (ab4729, abcam), 5 μg anti-H3K27me3 (07-449, Millipore) and 5 μg of normal rabbit IgG overnight at 4 °C. Antibody-protein complex was captured using 50 μl of pre-blocked Dynabeads A:G mixture for 3 h at 4 °C. Samples were processed further as described in the earlier section.

*Library preparation, sequencing, and data analysis.* Five nanograms of purified ChIP DNA was used for library preparation using NEB ultra II DNA kit (NEB, USA) as described in the earlier section. Sequencing reads (100 bp PE) were obtained on the HiseqX platform at Macrogen Inc, Korea. Sequencing reads were trimmed using TrimmomaticPE for Truseq2:PE adapters and reads with quality greater than phred 33 were retained. Quality of sequencing reads were determined using fastQC. High quality sequencing reads were aligned to the zebrafish danRer10 genome version using default parameters of BWA[82]. Aligned reads were subsampled to 40 million reads in each sample using BBMap[83]. RPKM normalized Input signal subtracted BigWig tracks for visualization were generated using the bamCompare tool from deepTools 3.3.2.

## ATAC-seq and data analysis

Single embryos at 14 ss from the incross of *satb2*[+/−] were hand dechorinated and harvested in 50 ul of ice chilled 1× DPBS (Invitrogen). Five microlitres of cell suspension was used for identifying the genotype of each embryo. Cell suspension from three embryos were pooled together and processed for Omni-ATAC seq[76] with modification from Amanda Ackermann lab. Briefly, cells were washed with 1× DPBS and resuspended in cell lysis buffer (10 mM Tris pH 7.5, 10 mM NaCl, 3 mM KCl, 0.1% NP-40, 0.1% Tween20 and 0.01% Digitonin) and

incubated on ice for 3 min. Further, cells were washed with a wash buffer (10 mM Tris pH 7.5, 10 mM NaCl, 3 mM KCl and 0.1% Tween20) by centrifugation at 500 xg for 10 min at 4 °C. Supernatant was discarded and the pellet was resuspended in 25 μl 2× Tagmentation buffer (Illumina, catalog #15027866), 16.5 μl DPBS, 0.5 μl 10% Tween 20, 0.5 μl 1% Digitonin, 5 μl nuclease and 2.5 μl Tn5 transposase enzyme (TDE1, Illumina, catalog # 15027865) and incubated for 28 min at 37 °C. After the tagmentation reaction, DNA was isolated using the Zymo DNA clean and concentrator kit (Zymo). Purified DNA was used as an input to generate a library by amplifying with 2× Q5 DNA polymerase mix (NEB) and indexing primers. Optimal cycles were determined using qPCR analysis. Amplified libraries were purified using Agencourt ampure XP beads to remove adapters and larger fragments.

For Satb2 overexpression studies, 200 pg 3xFLAG Satb2 were injected at 1 cell stage. Embryos (a pool of 10 embryos for each replicate) were harvested at 4.5 hpf and processed for ATACseq as described above.

Sequencing reads (41 bp PE) were obtained on NextSeq-550 at IISER Pune and trimmed for Nextera adapters using default parameters of Trimmomatic PE. Trimmed reads were aligned to danRer10 using default parameters of Bowtie2[95]. Briefly, BAM files were subsampled to 55 million reads in each sample using BBMap and sorted by name. Paired end bed files were obtained using the bamtobed function of bedtools. Reads were displaced by +4 bp and −5 bp. Peak calling was performed using MACS2 callpeak -f BEDPE -q 0.05 –nomodel –extsize 200 –gsize 1.3e9 –keep-dup 2 parameters. Consensus peaks were obtained using a custom R script used for ChIPseq analysis. BigWig files were generated using bamCoverage (deepTools). Peaks were annotated to the nearest gene using Homer and classified into promoter (±2 Kb) and non-promoter regions. K-means clustering was performed around ±2 Kb of Satb2 peak center using deepTools. Clusters were annotated using Homer and gene ontology analysis was performed using webGestalt. Motif analysis was performed using findMotifsGenome.pl from Homer.

## Differential chromatin accessibility analysis (Diffbind)

The differential chromatin accessibility analysis of *satb2* mutant and wild type embryos at 14 ss was performed using the DiffBind[96]. Significantly differentially accessible peaks were identified using the Deseq2 package and only sites with FDR < 0.05 and fold change of >Log2 (±1.5) were used for further analysis. Differential accessible sites were annotated to the nearest gene using Homer. Core promoter was defined as ±2 Kb from the TSS.

## Nucleosome occupancy analysis

Nucleosome occupancy analysis was carried out using the nucleoATAC suite with default parameters for *satb2* mutant and wild type ATACseq datasets. Nucleosome fuzziness scores were obtained and used for calculating the difference in nucleosome phasing upon loss of function of *satb2*.

## Single-cell RNAseq analysis

To generate single-cell gene expression datasets, a mosaic over expression system was generated by injecting one cell of 16 cell stage embryos with 200 pg of 3xFLAG-Satb2. Only one cell per embryo was injected randomly irrespective of its spatial arrangement. Embryos were raised at 28.5 °C, hand dechorinated and dissociated further in 1× DMEM-F12 (Sigma Aldrich, USA) + 125 mM EGTA by hand tapping. Cells were collected by centrifugation at 300 xg for 2 min and resuspended in 1× DPBS (Gibco) + 0.1% BSA. Cell suspension was passed through 70 μm Flowmi cell strainers (Sigma). Cell viability was estimated using CountessII automated cell counters and samples with more than 95% viability were processed further with 10X Genomics Chromium. Approximately 8000 cells were captured in GEMs. scRNA-seq libraries were prepared as described in the 10X Genomics manual (Single Cell 3′ Reagent Kits V3, User Guide PN-1000092). Immediately following GEM generation, reverse transcription reaction was carried out using thermal cycler (Eppendorf Mastercycler pro) by incubating at 53 °C for 45 min followed by denaturation at 85 °C for 5 min. Single-stranded cDNA was purified using DynaBeads MyOne Silane Beads (Thermo Fisher Scientific, USA). cDNA amplification was performed for 11 cycles with initial denaturation at 98 °C for 3 min followed by repeated cycles of 98 °C for 15 s, 63 °C for 20 s and 72 °C for 1 min. Final extension was performed at 72 °C for 1 min. cDNA quality was determined on a Bioanalyzer 2100 high sensitivity assay. cDNA was further fragmented, end repaired and A-tailed according to manufacturer's instructions. Before proceeding for adapter ligation, samples were purified using double-sided clean up protocol with SPRI bead (Beckman Coulter). Adapter ligated sample was subjected to amplification for 11 cycles using indexing primer from Chromium i7 Multiplex Kit (PN-120262). Amplified libraries were again subjected to double-side purification using SPRI and quantified using Qubit fluorometer and library size was estimated using Bioanalyzer 2100.

1.5 pM of the denatured libraries was used as an input to obtain sequencing reads using 28 cycles for read1, 8 cycles for indexes and 101 cycles for read2 on NextSeq 550 at IISER Pune.

Sequencing data were further processed with default parameters using cell Ranger 3.0.2 (10X Genomics). Sequencing reads were aligned to the danRer10 genome using STAR aligner. Feature matrices generated using cell Ranger were utilized for further analysis using Seurat 3.0.

## mRNA sequencing and differential gene expression analysis

For Satb2 overexpression studies, 3x-Flag-Satb2 mRNA was injected at 1 cell stage and embryos

were harvested at 4.5 hpf and were lysed in RNA iso plus (DSS-Takara, India) to extract total RNA. Each sample consisted of at least 10 embryos per experiment. For Satb2 knock-down studies, 4 ng morpholino mix was injected at 1 cell stage, and embryos were harvested at 4.5 hpf. Approximately 500 ng of Total RNA with RIN > 9 was used as a starting material to prepare mRNA sequencing libraries using the SENSE mRNA seq kit (Lexogen GMBH, Austria). All cDNA samples were amplified for 10–15 cycles depending on cycle number estimation by qPCR using the PCR add-on kit (Lexogen GMBH, Austria). Amplified libraries were purified using two rounds of 0.8x volume of Hi-prep PCR purification kit. The concentration of libraries was estimated using the Qubit 4 DNA HS system (Thermo Scientific). Finally, all the libraries were pooled and subjected to high throughput sequencing using 100 bp PE chemistry on Hiseq 2500 (Macrogen, Korea).

Sequencing reads were trimmed for quality using Trimmomatic and aligned to daRer10 genome assembly using HISAT2 aligner. Counts for each gene feature was estimated using the FeatureCounts package from Rsubread. Differential expression analysis was performed for replicates using EdgeR. A volcano plot for significantly differentially expressed genes was generated using the PlotVolcano tool from the Galaxy server. Gene ontology analysis for upregulated and downregulated genes were performed using a web-based tool, gProfiler.

**TF binding promoter scan**. Consensus binding sites for respective TFs were extracted from JASPAR database and MAST (MEME Suite) was used for scanning genomic regions extracted from UCSC server using the version danRer10.

**NT2/D1 cell culture**. Human embryonic carcinoma cell line NT2/D1 (NTERA2-clone D1) were a kind gift from Dr. Peter Andrews, University of Sheffield, UK. They were cultured in Dulbecco's Modified Eagle's Medium with sodium pyruvate, high glucose (DMEM, Sigma-Aldrich, St. Louis, Missouri, USA) supplemented with 10% fetal bovine serum (Invitrogen, Carlsbad, California, USA), 2 mM L-glutamine (Invitrogen, Carlsbad, California, USA) and penicillin-streptomycin (Invitrogen, Carlsbad, California, USA) and maintained at 37 °C under 5% $CO_2$ atmosphere. NT2/D1 cells were sub-cultured upon reaching 70% confluency by gentle scraping.

**Transfection of overexpression constructs in NT2/D1**. For overexpression of SATB2 in NT2/D1 cells, $0.8 \times 10^6$ cells were seeded in 60 mm cell culture grade plates (Corning, New York, USA). 16 to 20 h post-seeding, the cells were transfected with either empty FLAG vector or FLAG-SATB2 constructs for overexpression using Lipofectamine 2000 (Invitrogen, Carlsbad, California, USA) as per the manufacturer's guidelines.

**All-trans retinoic acid-mediated differentiation of NT2/D1**. All-trans-retinoic acid (RA) (Sigma-Aldrich, St. Louis, Missouri, USA) was reconstituted at a concentration of 5 mg/ml in DMSO (Sigma-Aldrich, St. Louis, Missouri, USA) and stored at −80 °C. For differentiation, NT2/D1 cells were harvested using 0.05% trypsin resuspended in fresh medium and seeded at a density $1 \times 10^6$ cells in 100 mm tissue culture dish (Corning, New York, USA). Cells were allowed to grow for 24 h following which $1 \times 10^7$ cells were harvested for RNA and protein extractions as 0 day control. RA was added to the remaining plates at a concentration of 13.7 µM for the rest of 5 days. Each day cells were either replenished with fresh medium and RA or harvested for RNA.

**qRT-PCR analysis**. Embryos were collected at the desired stage and were lysed in RNAiso plus (DSS-Takara, India) to extract total RNA. Each sample consisted of at least 10 embryos per experiment and all experiments were repeated at least three times independently. cDNA was synthesized using High-capacity cDNA synthesis kit (Applied Biosystems), following the manufacturer's instruction.

Quantitative real-time PCR was performed using SYBR green chemistry (KAPA biosystems) using gene-specific primers (Supplementary Table 2) below on ViiA7 thermal cycler (Applied Biosystems). Changes in threshold cycles were calculated by subtracting the Ct values of the gene of interest from that of housekeeping control (for qRT-PCR) [Ct (target genes) – Ct (ef1a or GAPDH)]. ΔCt values of specific target genes from the experimental samples were then subtracted from their respective control samples to generate ΔΔCt values. The fold changes were calculated using the formula: $2^{\wedge} (-\Delta\Delta Ct\ value)$.

**Protein extraction and immunoblotting**. To confirm the absence of Satb2 in zebrafish larvae. Larvae were manually dechorionated, deyolked (55 mM NaCl, 1.8 mM KCl, 1.25 mM NaHCO$_3$) and harvested at 48 hpf by boiling in 2× Laemmli buffer (0.25 M Tris-Cl pH 6.8, 1% SDS, 1% β-mercaptoethanol, 15% glycerol) at 98 °C for 5 min and electrophoresed on 7.5% SDS-PAGE gel. Proteins were transferred to 0.45 µM PVDF membrane (Millipore, USA) by wet transfer method at 0.6 A for 2.5 h at 4 °C. Non-specific sites on the membrane were blocked using 5% BSA and further incubated with inhouse anti-SATB2 antibody (1:500) in 0.3% BSA overnight at 4 °C with constant rocking. Next day, membrane was washed with 1× TBST (50 mM Tris-Cl pH 7.4, 150 mM NaCl, 0.1% Tween-20) four times for 7 min each and incubated with anti-rabbit HRP conjugated antibody (1:10000

STAR124P, Bio Rad, USA) in 1× TBST for 45 min at RT. After removing excess secondary antibodies by repeated washes with 1× TBST, signal was developed with the Clarity western ECL substrate (Bio Rad, USA) and captured with ImageQuant LAS 4000 system (GE Healthcare, USA).

For experiments with NT2/D1 cells pellets were resuspended in RIPA buffer (10 mM Tris (pH 8.0), 1 mM EDTA (pH 8.0), 0.5 mM EGTA, 1% Triton X-100, 0.1% sodium deoxycholate, 0.1% SDS, 140 mM NaCl) containing 1× PIC and lysed by repeated freeze-thaw cycles. The lysates were centrifuged at $20,000 \times g$, 4 °C, 30 min to eliminate the cellular debris. The supernatant was collected in a fresh microfuge tube. The concentrations of protein were determined by performing BCA assay (Thermo Fisher Scientific, USA). Equal amounts of protein lysates were boiled in 1× Laemmli buffer (0.5 M Tris-HCl pH 6.8, 28% glycerol, 9% SDS, 5% 2-mercaptoethanol, 0.01% bromophenol blue) for 10 min and subjected to electrophoresis on a 7.5% SDS-PAGE gel. The separated proteins were transferred onto PVDF membrane (Millipore, USA) using a phosphate-based transfer buffer (10 mM sodium phosphate monobasic, 10 mM sodium phosphate dibasic) at 4 °C, 600 mA, 2 h. After the completion of transfer, membranes were blocked in 5% skimmed milk, incubated overnight at 4 °C with the anti-FLAG antibody prepared in 5% BSA. The membranes were washed thrice with the buffer containing 20 mM Tris buffer pH 7.4, 500 mM NaCl and 0.1% tween 20 (TST) the next day and incubated with the appropriate secondary antibodies conjugated with horseradish peroxidase for an hour at RT. Following this, the membranes were again washed thrice with TST buffer. The blots were developed using Immobilon Western Chemiluminescent HRP Substrate (Millipore, USA) and detected using ImageQuant LAS 4000 (GE Healthcare, USA) according to the manufacturer's instructions.

**Statistical analysis**. GraphPad Prism (GraphPad software, USA) was used for statistical analysis. The type of analysis used in each experiment is mentioned in the respective figure legends.

**Reporting summary**. Further information on research design is available in the Nature Research Reporting Summary linked to this article.

## Data availability
All the raw data generated in this study, including RNA-seq, ATAC-seq, ChIP-seq and scRNAseq data can be accessed on GEO [GSE183002]. All other relevant source data supporting the key findings of this study are provided with this paper. Source data are provided with this paper.

## Code availability
All the custom computational codes used in this study are available on Zenodo https://doi.org/10.5281/zenodo.5336938 and upon request from the first (saurabh.j.pradhan@gmail.com) or the corresponding author (sanjeev@iiserpune.ac.in). Source data are provided with this paper.

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

## Acknowledgements

We are grateful to the members of C.-P.H. and SG lab for discussions. Authors thank Shubha Tole for providing embryonic mouse tissues. Authors are grateful to Alessandro Mongera and Chetana Sachidanandan for generous help with Tg: Sox10: GFP line. Authors would like to thank Satyajeet Khare, Vanessa Barone, Jyothish S., Shalini Mishra, Yoshita Bhide, and Keshav Jha for assistance in experiments. We would also like to thank Chaitanya Dingare for valuable suggestions. We thank Diana Pinhiero and Alexandra Schauer for critical reading of early versions of the manuscript. This work was supported by the Centre of Excellence in Epigenetics program of the Department of Biotechnology, Government of India Phase I (BT/01/COE/09/07) to S.G. and R.K.M., and Phase II (BT/COE/34/SP17426/2016) to S.G. and JC Bose Fellowship (JCB/2019/000013) from Science and Engineering Research Board, Government of India to S.G., DST-BMWF Indo-Austrian bilateral program grant to S.G. and C.-P.H. The work using animal models was partly supported by the infrastructure support grants from the Department of Biotechnology (National Facility for Laboratory Model Organisms: BT/INF/22/SP17358/2016 and Establishment of a Pune Biotech Cluster, Model Organism to Human Disease:

B-2 Whole Animal Imaging & Tissue Processing FacilityBT/Pune-Biocluster/01/2015). S.J.P. was supported by Fellowship from the Council of Scientific and Industrial Research, India and travel fellowship from the Company of Biologists, UK. P.C.R. was supported by the Early Career Fellowship of the Wellcome Trust-DBT India Alliance (IA/E/16/1/503057). A.S. was supported by UGC and R.S. was supported by CSIR India. M.S. was supported by core funding from the Tata Institute of Fundamental Research (TIFR 12P-121).

## Author contributions

S.J.P. conceived the study, designed experiments, interpreted data, generated CRISPR mutants, performed all zebrafish embryology experiments with assistance from other authors, performed RNA-seq, ChIP-seq, ATAC-seq, scRNA-seq and associated data analysis, generated figures, and wrote the manuscript. P.C.R. performed phylogenetic analysis, protein domain architecture analysis, and performed bulk RNA-seq analysis. M.Sm. designed and assisted in overexpression experiments. A.S. performed micro-CT scans, assisted in scRNA-seq and ATAC-seq. K.S. cloned 3xFLAG-Satb2 and assisted in experiments. M.S.O. performed alcian blue staining and WISH. R.S. performed SATB2 overexpression experiments in NT2/D1. M.P. assisted in stage-dependent ChIP-seq experiments. O.D. performed WISH and morpholino mediated knockdown. Y.T. generated plots for Satb2 OE mRNA-seq analysis. R.K.M. contributed resources. G. Ds. provided assistance in experiments with Tg:Sox10 transgenic line. G.D. edited the manuscript and provided expertise in evolutionary perspectives. A.J.G. designed GLT experiments and provided expertise in pou5f3 mutants and ZGA regulation. M.S. provided expertise in zebrafish experiments, design of morpholino experiments and provided resources. C.-P.H. designed experiments, interpreted data, provided resources, edited manuscript, obtained funding, and supervised the study. S.G. conceived the project, designed experiments, interpreted data, wrote manuscript, obtained funding and other resources, and supervised the entire study. All authors read and approved the final manuscript.

## Competing interests

The authors declare no competing interests.

## Ethical approval

All the experimental procedures using zebrafish were carried out in accordance with the guidelines from the institutional animal ethics committee at IISER Pune and IST Austria. Experiments with mice models (*Mus musculus*) were carried out at TIFR, Mumbai adhering to Institutional ethics guidelines.
