## [Peer Review File · Nature Communications]

Satb2 acts as a gatekeeper for major developmental transitions during early vertebrate embryogenesisReviewers' Comments:

Reviewer #1:

Remarks to the Author:

The article by Pradhan and colleagues investigates temporal differences in Satb2 function during zebrafish development. The authors focus on early roles for Satb2 that would likely be performed by maternally contributed Satb2 versus later (zygotic) roles for Satb2. They present a large number of sequencing datasets which they use to argue that Satb2 has two distinct roles- one as a repressor in early development, and as an activator later in development. They also suggest that Satb2 may somehow be involved in zygotic genome activation.

A positive of the work described in the manuscript is the amount of sequencing data presented. However, the conclusions drawn from this data are not always consistent within the present analysis and also diverge with what has been shown in other systems (mostly mouse) about Satb2. The primary conclusion at the end of the results, that contrasting stage-dependent function of Satb2 is a consequence of qualitatively distinct binding, has already been shown in mouse- further it is not stage-dependent, but rather cell-type dependent. This key point is related to a first weakness of the manuscript, which is that the authors have chosen to perform this study in zebrafish, but have not utilized key advantages of this model system, such as fluorescent reporters that would allow them to sort specific cell types and really refine cell type specific behavior of satb2. Such data could provide an important advance in our knowledge of satb2. Instead they have done large scale tissue evaluations of satb2 function which have the effect of averaging or blending the role of satb2 in these cells.

The authors emphasize the maternal-zygotic transition and argue for a role for satb2 in this transition. But it's not clear what data support this role. The authors have simply shown, at best, that satb2 has a cell type specific function in early, cleavage stage embryos. The fact that satb2 needs to be down-regulated as development moves to different stages of differentiation does not mean that it is required for a maternal-zygotic transition, unless this is a unique feature of zebrafish development. This point relates to a second weakness of this paper, which is the absence of clear comparisons to data published in mouse. This is particularly relevant to the role of Satb2 in early development. In mouse, satb2 is expressed in the blastocyst where it positively regulates Nanog (Savarese et al. 2009), in contrast to the specific repressive role the authors argue for satb2 in zebrafish. It is also not clear what such a specific zygotic transition role would mean for mammals where zygotic gene expression begins in one-to-eight-celled embryos. In mouse, satb2's function in early blastocyst cell fate determination occurs throughout this time from one-celled to hundreds of cells (Goolam & Zernicka-Goetz 2017).

Other major considerations:

1. In Figure 1E, the authors show survival data for their genotypes used in this study. WT individuals have a very low (25% initial survival rate). Why is survival rate so low among WT fish, and how can the percentage surviving animals go up over time? There is something not right with these data, and/or something not right with the animals used in this study.

2. The authors suggest that "the presence of both SATB1 and SATB2 is only apparent in jawed fish and later species, suggesting these proteins presumably co-evolved with the evolution of the jaw structure." Co-evolution implies an interaction, like predator-prey co-evolution. It is not clear what the authors are trying to communicate in this section of the paper. Further, why would gene duplication after the split with agnathans necessarily imply an association with the jaw as opposed to any other structure that is derived in the stem gnathostome lineage, such as bone or teeth, both of which satb2 has known developmental roles?

3. The authors have broadly placed Satb2 into two categories- maternal and zygotic in a way that implies that they are different proteins rather than the same protein acting in different cells at

different times. This causes several downstream problems. First, the analysis of "zygotic" satb2 does not take into account what is already known about cell-type specific behavior of satb2- it is expressed in multiple cell types during development- neural crest, but importantly only a subset of neural crest, neurons, but again only a subset of neurons, and osteoblasts. The analyses on zygotic satb2 lumps all of these together as if satb2 is doing the same thing in these cells, interacting with the same partners, etc. This has been shown in a number of studies in mice not to be the case. Even in the case of neurons, satb2 is doing very different things during cortical development compared to its function in adult neurons. So the idea that zygotic Satb2 is simply a gene expression activator is very oversimplified. That said, the expression data shown are not controversial data- previous work has shown an involvement of Satb2 in regulating skeletal genes and some NC genes. However, their conclusion that Satb2 is required for early stages of NC development is not substantiated since they can't isolate this cell type in their analysis. Also, if this is true, it would be specific to zebrafish since Satb2 is not expressed in pre-migratory NC in mice. This key difference between zebrafish and mammals is not discussed.

4. on page 10, the authors suggest "that unlike zygotic Satb2, the maternal pool of Satb2 is involved in transcriptional repression and possibly regulates the same group of genes in an opposite manner." They have the chip-seq data which could address this claim. They do use this data later to argue that satb2 is binding different regions maternally vs zygotically (fig. 7a). So both of these statements can't be true.

4. The authors employ gene expression analyses at 80% epiboly, 5-6 ss and 14 ss, in order to study NC development. First, it has previously been shown that Satb2 is widely, if not ubiquitously, expressed in the early zebrafish embryo (Ahn et al. 2010), so these data do not specifically address NC. As noted above, it is not clear why lineage markers would not be used to provide more specific analyses of cell type behavior- the authors appear to assume that satb2 is doing the same thing in all cells in this zygotic period.

5. Further, and more importantly, the expression data reported by Ahn et al. 2010 are not consistent with the biphasic model of satb2 expression reported here. The authors do not discuss this point.

Minor considerations:

Fig. 1, panel C- why does mutant band run lower in KO vs het?

Reviewer #3:

Remarks to the Author:

In this manuscript, Pradhan et al. analyzed Satb2 localization and function in zebrafish and other model systems. They first investigated the knockout phenotype in zebrafish and found craniofacial defects as observed in mammals, implying that analyzing the Satb2 function in zebrafish is relevant to human disease. They then went into genome-wide analysis, including, RNA-seq, ChIP-seq, and ATAC-seq, which suggested that the zygotic knockout phenotype results from dysregulation of gene expression through altered chromatin structure. The aberrant expression of some key genes was confirmed by in situ hybridization. To analyze the function of maternal Satb2, its overexpression was employed. The overexpression induced chromatin silencing rather than activation. The differential effects of Satb2 in the zygotic genome activation and in differentiation may be explained by the partner transcription factor. The function of biphasic expression of Satb2 was thus explained.

The study was well performed and thorough, providing significant advance in understanding Satb2 function during development and differentiation. Furthermore, this study provokes our thinking how a chromatin protein can bimodally regulate gene expression. The conclusions were well-supported by

the high-quality data presented.

I have a few comments that may improve the manuscript.

1. There is no consideration for the splice variants of *Satb2*, while the existence of variants is suggested when generating the phylogenetic tree. Could the authors exclude a possibility that maternal and zygotic *Satb2* protein differ in the function due to alternative splicing and/or alternative translation initiation? This question can be experimentally addressed, but at least, the authors should more look into the database and comment on.
2. Fig. 2a and Fig. 4a. The genomic feature of the classification should be shown along with the ChIP-seq and ATAC-seq data, because it is not clear if one category is indeed enriched or just appears proportionally to its abundance in the genome.
3. Fig. 2e. I understand that the blue circle indicates *Satb2* bound genes. Why the numbers differ in the two (left and right for upregulated and downregulated genes in the KO, respectively)?
4. Supplementary Fig. 3, 5, and 7. Western blotting panels require the positions of size standards.

Hiroshi Kimura

Reviewer response NCOMMS-20-39460-T

Reviewer #1 (Remarks to the Author):

The article by Pradhan and colleagues investigates temporal differences in Satb2 function during zebrafish development. The authors focus on early roles for Satb2 that would likely be performed by maternally contributed Satb2 versus later (zygotic) roles for Satb2. They present a large number of sequencing datasets which they use to argue that Satb2 has two distinct roles- one as a repressor in early development, and as an activator later in development. They also suggest that Satb2 may somehow be involved in zygotic genome activation.

A positive of the work described in the manuscript is the amount of sequencing data presented. However, the conclusions drawn from this data are not always consistent within the present analysis and also diverge with what has been shown in other systems (mostly mouse) about Satb2. The primary conclusion at the end of the results, that the contrasting stage-dependent function of Satb2 is a consequence of qualitatively distinct binding, has already been shown in mouse- further it is not stage-dependent, but rather cell-type dependent. This key point is related to a first weakness of the manuscript, which is that the authors have chosen to perform this study in zebrafish but have not utilized key advantages of this model system, such as fluorescent reporters that would allow them to sort specific cell types and really refine cell type specific behaviour of satb2. Such data could provide an important advance in our knowledge of satb2. Instead, they have done large scale tissue evaluations of satb2 function which have the effect of averaging or blending the role of satb2 in these cells.

Response: We thank the reviewer for the thoughtful critique. Indeed, cell type specificity and organismal context are two important factors that need to be considered while interpreting our data and we have tried to do so in the revised version.

Briefly, we sought to use zebrafish as a model as it allowed us to probe the function of Satb2 throughout the developmental progression, from a single cell to an adult. From the outset, we hoped to uncover either completely novel or context-specific functions of Satb2 and how it is deployed temporally to achieve distinct outcomes. As the reviewer pointed out, we also hoped to uncover its cell-type specific activities. We believe that the analysis presented in this manuscript has successfully touched upon most of these issues.

We were keen to glean evolutionary insights using zebrafish as an early vertebrate model that mimics Satb2 associated craniofacial abnormalities first reported in higher mammals. This is important since Satb1 and Satb2 likely diverged with the evolution of jawed fish.

In this context, it is also noteworthy that defined cell types emerge only after early cleavage stages of zebrafish embryogenesis, and thus, we do not attribute the early embryonic function of Satb2 to any specific cell type. This aspect may be unique to zebrafish Satb2 and may differ as opposed to the mouse version of the protein making a direct comparison untenable. We thank the reviewer for raising this important issue and we have tried to account for this distinction in the revised manuscript.

One important novel conclusion that emerged from our studies concerns the functional distinction i.e., bimodality between the maternally deposited and zygotically synthesized Satb2. As described in detail, this bi-modal behavior is pertinent to the early stages of embryonic development.

As pointed out by the reviewer, during later stages, Satb2 likely exhibits cell type specific functions. As alluded to by the reviewer, we considered performing functional studies in the background of a transgenic reporter line. However, due to the nature of the Satb2 mutation, we were unable to obtain homozygous *satb2*^{-/-} fish. Owing to this technical challenge, we could not identify homozygous mutants before subjecting them to FACS sorting necessary to perform the cell type specific analysis. Consequently, throughout the manuscript, we have used embryos obtained by crossing heterozygous parents. The genetic markers which we have characterized in the absence of Satb2 are specific to early neurogenesis and neural crest development (confirmed by previous studies reported in literature and using gene ontology analysis).

To substantiate our conclusions, we have now performed single-cell gene expression analysis in wild type vs mutant embryos. This has allowed us to identify cell types that show significant changes upon Satb2 deletion. Consistent with our previous analysis, scRNAseq data shows maximum reduction in neural progenitors and neural crest cell populations arguing for a tissue specific function. We also observe an increase in the endothelial population and lateral plate mesoderm population. However, we only analyzed early stages of embryos and thus, are not able to comment on effects of loss of function of Satb2 on neural sub-types and future studies will focus on analysis of later stage embryos. Importantly, to analyze effect at later stages of neural development and cell-subtypes experiments, mouse models would prove to be highly relevant.

Throughout our current study, our primary focus was to assess the function of Satb2 during early stages of neurogenesis including its onset. As a result, zebrafish was our system of choice to track neural development during the early developmental time window. Nevertheless, it will be valuable to integrate our findings with the conclusions derived from other model systems to evolve a more complete picture of the process with

a deeper understanding of mechanistic underpinnings underlying neural crest specification and craniofacial development.

As suggested by the reviewer, we also utilized Sox10: GFP reporter lines to analyze the neural crest specific function of Satb2. First, we evaluated and compared the expression of Satb2 in cranial neural crest and non-neural crest cells. Expectedly, we observed significant enrichment in expression levels of Satb2 in the cranial neural crest cells. These results corroborate the transcriptome analysis in crestin: GFP cells which serves as a marker for cranial neural crest cells.

Next, we isolated cranial neural crest and non-neural crest cells using Sox10: GFP transgenic fish line by FACS and performed ChIP-seq for Satb2. We observed significant correlation between datasets generated using sorted population and whole embryos tissue level analysis suggesting that during these stages, both approaches provided similar insights and we can use whole embryo approach at earlier stages when reporter expression is below the detection limit of microscope and flow cytometer.

The authors emphasize the maternal-zygotic transition and argue for a role for satb2 in this transition. But it's not clear what data support this role. The authors have simply shown, at best, that satb2 has a cell type specific function in early, cleavage stage embryos. The fact that satb2 needs to be down-regulated as development moves to different stages of differentiation does not mean that it is required for a maternal-zygotic transition unless this is a unique feature of zebrafish development. This point relates to the second weakness of this paper, which is the absence of clear comparisons to data published in the mouse. This is particularly relevant to the role of Satb2 in early development. In mouse, satb2 is expressed in the blastocyst where it positively regulates Nanog (Savarese et al. 2009), in contrast to the specific repressive role the authors argue for satb2 in zebrafish. It is also not clear what such a specific zygotic transition role would mean for mammals where zygotic gene expression begins in one-to-eight-celled embryos. In mouse, satb2's function in early blastocyst cell fate determination occurs throughout this time from one-celled to hundreds of cells (Goolam & Zernicka-Goetz 2017).

Response: We thank the reviewer for this astute insight. Indeed, the onset of ZGA is a unique feature of zebrafish and thus cannot be directly compared to mammalian ZGA with respect to its timing and cellular changes. However, molecular mechanisms which regulate the process of ZGA are highly conserved between zebrafish and humans (Jukam *et al.* Dev Cell 2017; Schulz and Harrison Nat. Rev Genet. 2019). Importantly, few studies suggest that regulation of ZGA by OCT4 is a conserved mechanism between zebrafish

and humans while mice are the outliers in this regard (Lee *et al.* Nature 2017; Leichsenring *et al.* Science 2013; Gao *et al.* Cell 2018).

We appreciate reviewers' concern that downregulation of *satb2* does not necessarily imply that it is required for a maternal-zygotic transition. Indeed, to assess this possibility, we had engineered continuous expression of Satb2 by performing overexpression experiments and our data show that sustained expression of Satb2 during early developmental stages impairs activation of zygotic genes significantly. The effect seems specific in nature as retention of Satb2 did not affect the maternally deposited genes.

It should be noted that we do not comment on any cell type specific activity of Satb2 during ZGA, as zebrafish embryos contain only two cell types at this stage: enveloping layer (EVL) and deep cells. The single-cell analysis was employed in this context, simply to highlight cell-autonomous mode of regulation by Satb2 i.e., we did not intend to highlight cell type specific activity of Satb2.

However, our scRNAseq data on embryos expressing higher levels of Satb2 in mosaic manner suggests negative correlation between Satb2 and zygotic genes. In our view, these data strongly argue that in zebrafish, downregulation of Satb2 is necessary for efficient progression of ZGA. We cannot comment on zygotic genome activation in mammals vis-a-vis Satb2 loss during ZGA. It is important to note that Goolam & Zernicka-Goetz 2017 did not specifically categorize global effects on maternal versus zygotic genes upon overexpression of SATB2. However, it would be interesting to follow up on this line in the future and we have now discussed this possibility in the revised version of the manuscript (Line numbers 514-524).

Savarese *et al.* (2009) have suggested positive regulation of Nanog by Satb2 in the early blastocyst, However, as the reviewer correctly pointed out our data do not demonstrate that in zebrafish, Satb2 downregulates Nanog. In fact, Nanog expression levels are not perturbed upon over-expression of Satb2. This could be due to species-specific differences in Satb2 activities, and we have discussed this in the revised version of the manuscript (Line numbers 544-548). Our data show that, Satb2 downregulates Pou5f3 but upregulates Sox19b. Together these data show that the effect of *satb2* on pluripotency factors is qualitatively non-uniform and we hypothesize that ZGA is affected due to such homeostatic imbalances resulting from overexpression of Satb2.

As suggested by the reviewer, we have included a section in the discussion about comparison between studies in mice and this analysis (line numbers 541-548).

The study cited by the reviewer (Goolam & Zernicka-Goetz, 2017) is focused primarily on *Satb1*, another member of the protein family and as such cannot be used for a direct comparison. In fact, these authors mention that *Satb2* knockdown does not influence embryonic development adversely presumably due to activity of residual *Nanog* expression. However, it should be noted that knockdown of *Satb2* partly restores gene expression programs which are perturbed upon the knockdown of *Satb1*. Thus, highlighting antagonism between mouse homologs of *Satb1* and *Satb2*. We have not observed such antagonism between these two proteins at early stages of zebrafish development. and we have discussed this possible species-specific distinction in the revised version of the manuscript (Line numbers 514-524).

In sum, it will be interesting to elaborate on these studies to analyze the effect of *Satb2* in other mammalian systems to assess possible limits of conservation across vertebrate species. However, our data and other studies suggest conserved function for *Satb2* in neural crest regulation and craniofacial development.

Other major considerations:

1. In Figure 1E, the authors show survival data for their genotypes used in this study. WT individuals have a very low (25% initial survival rate). Why is the survival rate so low among WT fish, and how can the percentage of surviving animals go up over time? There is something not right with these data, and/or something not right with the animals used in this study.

Response: Data represented in this figure summarizes the observed frequency of animals with the corresponding genotype in a test population. For example, on a given day before day 14, if we screen 48 larvae from the in-cross of *satb2*^{+/-}, we observe 25% WT (~ 12), 50% heterozygous mutants (~ 24) and 25% mutant embryos (~ 12) in a clutch in accordance with the Mendelian ratio. Similarly, on day 21, we do not observe any homozygous mutant surviving in a population (or very rare population which we have analyzed using microCT) and thus percent frequencies of Wild type and heterozygous mutants increased to be around 50%. We have now changed the Y-axis label to Survival rate (percent frequency). We hope this provides desired clarification as per the reviewer's request.

2. The authors suggest that “the presence of both SATB1 and SATB2 is only apparent in jawed fish and later species, suggesting these proteins presumably co-evolved with the evolution of the jaw structure.” Co-evolution implies an interaction, like predator-prey co-evolution. It is not clear what the authors are trying to communicate in this section of the

paper. Further, why would gene duplication after the split with agnathans necessarily imply any association with the jaw as opposed to any other structure that is derived in the stem gnathostome lineage, such as bone or teeth, both of which *satb2* has known developmental roles?

Response: Here, by using the term ‘co-evolution’, we implied that the ancestral form of *Satb* protein has diverged into two distinct proteins- *Satb1* and *Satb2* with the evolution of jawed fish. Previous studies have proposed a specific role for *SATB2* during determination of vertebrate jaw structures including teeth (Fish *et al.*, *Evol dev* 2011; Dobрева *et al.*, *Cell* 2006; Sheehan-Rooney *et al.*, *Dev Dyn* 2013). Therefore, we used the term ‘co-evolution’ to indicate the evolutionary dependence of jaw structures on *SATB2*. However, we agree with the reviewer that this does not mean functional interaction between the two and further evidence is required to argue for co-evolution. Therefore, now we use the specific term ‘divergence’. We also appreciate the point of involvement in bone and teeth development. Neural crest cells play an important role in tooth development and skeletal structures, specifically cranial bone development (Yamazak and Hayashi 2004; Mishina and Snider 2014). We have now modified the text to discuss the possible role of *Satb2* in regulating these tissue types as reported in mammals (line numbers 103-105). However, in the current study, we have used jaw development as one of the paradigms to highlight functional conservation and contribution of *Satb2* during neural crest specification.

3. The authors have broadly placed *Satb2* into two categories- maternal and zygotic in a way that implies that they are different proteins rather than the same protein acting in different cells at different times. This causes several downstream problems. First, the analysis of “zygotic” *satb2* does not take into account what is already known about cell-type specific behavior of *satb2*- it is expressed in multiple cell types during development- neural crest, but importantly only a subset of neural crest, neurons, but again only a subset of neurons, and osteoblasts. The analyses on zygotic *satb2* lump all of these together as if *satb2* is doing the same thing in these cells, interacting with the same partners, etc. This has been shown in a number of studies in mice not to be the case. Even in the case of neurons, *satb2* is doing very different things during cortical development compared to its function in adult neurons. So the idea that zygotic *Satb2* is simply a gene expression activator is very over-simplified. That said, the expression data shown are not controversial data- previous work has shown the involvement of *Satb2* in regulating skeletal genes and some NC genes. However, their conclusion that *Satb2* is required for early stages of NC development is not substantiated since they can’t isolate this cell type in their analysis. Also, if this is true, it would be specific to zebrafish since *Satb2* is not expressed in pre-migratory NC in mice. This key difference between zebrafish and mammals is not discussed.

Response: This was perhaps erroneously represented in the earlier version of the manuscript. We did not intend to suggest that maternal and zygotic *satb2* are different protein isoforms. The distinction was only meant to be temporal.

1. maternally deposited protein pool, and 2. zygotically synthesized pool.

We support this by showing that *Satb2* mRNA levels decrease significantly after the onset of ZGA. This suggests that maternally deposited *Satb2* mRNA is depleted from the system. To establish this unequivocally we show that only one form of *Satb2* is expressed throughout early embryogenesis and rule out the possibility of alternate splicing.

We appreciate the point of the reviewer that our analysis on bulk embryos could miss out on the information about cell type-specific regulation by *Satb2* as shown during mouse brain development. However, the purpose of these experiments is to probe how early during the developing embryo, genes that are responsible for neural crest development are positively regulated by *Satb2*. This analysis has helped us identify the critical time window of functional regulation by *Satb2* crucial to facilitate craniofacial development. Ideally, as the reviewer suggested, this analysis should be performed on isolated neural crest cells from the *satb2* mutants. This is, however, not possible due to the nature of *Satb2* mutation, as homozygous *Satb2*^{-/-} fish can't be easily retrieved.

To circumvent this problem, in the current version of the manuscript, we have included analysis using scRNAseq in WT and mutants. We show specific effect on neural progenitors and neural crest cells (Main Fig. 2). Moreover, *Satb2* ChIP-seq analysis on Sox10pos cells also highlights neural crest-specific activity of *Satb2* (Main Fig. 3). We are sensitive to the fact that these data do not allow us to probe into the role of *satb2* during specification of neuronal subtypes. Future studies will be necessary to address this question.

A recent study by Fazio *et al.* (Biorxiv <https://doi.org/10.1101/2020.11.01.364406>) reported a knockdown with morpholinos against *Satb2*. Similar KD approach was used by Ahn *et al.* 2010 to show that loss of *Satb2* results in reduced *sox10* expression (assessed by *sox10:gfp* reporter line) and craniofacial defects. However, it does not affect melanocyte development. Together these data suggest that *Satb2* has a specific function in cranial neural crest cells and not in neural crest cells which contribute to peripheral tissue types.

Figure 1. Data from Fazio et al. 2020 suggesting loss of function of Satb2 using previously characterized morpholinos leads to defective cranial skeleton (marked by Sox10:mCherry) but does not affect melanin pigmentation.

We also do not see the loss of pigmentation in the survivor Satb2 mutants (Supplementary Fig1) essentially supporting conclusions from this study. Moreover, in our manuscript, we address this question and subsequently support this notion by showing specific enrichment of Satb2 in head tissue of developing mouse embryos at (E9.5). We show higher enrichment in case of head tissue (source of cranial neural crest cells than trunk tissue). New analysis in Sox10^{pos} cells also suggest higher enrichment in cranial neural crest cells than non-neural crest cells in zebrafish.

Comments from the reviewer further suggest that Satb2 function in early neural progenitors could be restricted to zebrafish since Satb2 is not expressed in murine neural crest progenitors. However, RNA-seq studies from the laboratory of Filippo Rijj (Minoux *et al.* Science 2017) suggests significant expression of Satb2 in neural crest progenitors at neural crest subtypes from as early as E8.5.

Figure 2. Bar plot representing SATB2 expression in isolated neural progenitors and neural crest subtypes from developing mouse embryos.

Moreover, scRNAseq analysis of cranial neural crest cells (E8.5) and trunk neural crest cells (E9.5) also suggest enrichment of *Satb2* in cranial neural crest population compared to neural crest contributing to peripheral tissues (Soldatov *et al.* Science, 2018). This data also supports our ChIP-seq studies in mouse.

Figure 3. Expression of *Satb2* from scRNAseq dataset (Red: positive to blue: negative expression) <http://pklab.med.harvard.edu/ruslan/neural.crest.html>

4. on page 10, the authors suggest "that unlike zygotic *Satb2*, the maternal pool of *Satb2* is involved in transcriptional repression and possibly regulates the same group of genes in an opposite manner." They have the chip-seq data which could address this claim. They do use this data later to argue that *satb2* is binding different regions maternally vs zygotically (fig. 7a). So, both of these statements can't be true.

Response: In Figure 7A, we demonstrate that Satb2 binds in a differential manner at distinct stages (intergenic versus promoter regions). However, the nearest genes are the same. This is an interesting point and we have included this in the results/discussion sections (line numbers) suggesting not only differential interacting partners but also binding at different genomic regions potentially contributing to Satb2's contrasting functions. For example, we observe that Satb2 binds only intergenic regions of neural crest marker genes such as *sox10* at 512 and dome stage. This pattern of occupancy changes during later stages as Satb2 also binds the promoter region of *sox10*.

4. The authors employ gene expression analyses at 80% epiboly, 5-6 ss and 14 ss, in order to study NC development. First, it has previously been shown that Satb2 is widely, if not ubiquitously, expressed in the early zebrafish embryo (Ahn et al. 2010), so these data do not specifically address NC. As noted above, it is not clear why lineage markers would not be used to provide more specific analyses of cell type behavior- the authors appear to assume that *satb2* is doing the same thing in all cells in this zygotic period.

Response: Ahn *et al.* performed 'in situ' analysis which did not include stages of neural crest specification (6 somites) and NC differentiation (14 somites) in their analysis. In contrast, quantitative RNAseq analysis has provided detailed insights into the expression pattern of Satb2.

Fazio *et al.* (Biorxiv <https://doi.org/10.1101/2020.11.01.364406>) also analyzed previously published data to show that *satb2* is highly expressed in a migrating *crestin:EGFP+* NC cells sorted from zebrafish embryos at the 15 somite stage (Kaufman et al., 2016). As mentioned earlier, the dataset from Ahn *et al.* (2010) does not include this stage. Based on the RNA seq study performed by Kaufman *et al.*, it is clear that during early stages of neural crest development, Satb2 plays a cell type (NC) specific function.

**Zebrafish Neural Crest Tg(*crestin:EGFP*) sorted cells
(Kaufman et al. 2016)**

	crestin - (FPKM)	crestin + (FPKM)	Log2 Fold Change
satb2	0.775	13.597	4.134

(Fazio et al. 2020)

Figure 4. FPKM values for Satb2 in neural crest and non-neural crest cells (Kaufman et al. 2016).

We have validated observations from RNA-seq studies using qPCR utilizing another reporter line for marking neural crest population- Sox10: GFP. We isolated RNA from Sox10+ and Sox10- cells at 14 somites stage and observed that *Satb2* is indeed enriched in the neural crest population than in the non-neural crest thus further strengthening our conclusions (Supplementary Fig. 2a).

5. Further, and more importantly, the expression data reported by Ahn et al. 2010 are not consistent with the biphasic model of *satb2* expression reported here. The authors do not discuss this point.

Response: We thank the reviewer for pointing out this oversight and we have now discussed this point in the revised manuscript.

It should be noted that the expression data provided by Ahn *et al.* 2010 is non-quantitative WISH analysis. We have used high depth stage-dependent transcriptome data which allowed us to confirm and extend the biphasic nature of *Satb2* during early embryonic development. We analyzed datasets from two independent studies (Pauli *et al.*, Genome Res 2012 and EBI). Moreover, our analysis of the human stem cell differentiation model also supports this notion. This point is also discussed in the revised version (Line number 483-485).

Minor considerations:

Fig. 1, panel C- why does mutant band run lower in KO vs het?

Response: We appreciate this concern. This anomaly could be due to specific issues with gel electrophoresis run. We have now replaced this panel with a new plot. High-resolution melt curve analysis confirms that the product size is the same in heterozygous and homozygous mutants (assessed by T_m) confirming our conclusions.

Reviewer #3 (Remarks to the Author):

In this manuscript, Pradhan et al. analyzed Satb2 localization and function in zebrafish and other model systems. They first investigated the knockout phenotype in zebrafish and found craniofacial defects as observed in mammals, implying that analyzing the Satb2 function in zebrafish is relevant to human disease. They then went into genome-wide analysis, including, RNA-seq, ChIP-seq, and ATAC-seq, which suggested that the zygotic knockout phenotype results from dysregulation of gene expression through altered chromatin structure. The aberrant expression of some key genes was confirmed by in situ hybridizations. To analyze the function of maternal Satb2, its overexpression was employed. The overexpression induced chromatin silencing rather than activation. The differential effects of Satb2 in the zygotic genome activation and in differentiation may be explained by the partner transcription factor. The function of biphasic expression of Satb2 was thus explained.

The study was well performed and thorough, providing significant advance in understanding Satb2 function during development and differentiation. Furthermore, this study provokes our thinking about how a chromatin protein can bimodally regulate gene expression. The conclusions were well-supported by the high-quality data presented.

I have a few comments that may improve the manuscript.

1. There is no consideration for the splice variants of Satb2, while the existence of variants is suggested when generating the phylogenetic tree. Could the authors exclude a possibility that maternal and zygotic Satb2 protein differ in function due to alternative splicing and/or alternative translation initiation? This question can be experimentally addressed, but at least, the authors should look into the database more and comment on it.

Response: During phylogenetic tree construction, we excluded the splice variants of SATB1 and we did not find any splice variants for SATB2. Additionally, RNA sequencing data also confirms the absence of detectable splice variants for SATB2 in Zebrafish. To improve the clarity, the methodology section has been modified. We have specifically included the following sentence to clarify this point “SATB1 splice variants were removed, and no splice variants were observed for SATB2.”

To further validate that there are no splice variants expressed during these stages of zebrafish development, we analyzed Ribo-Seq data from Giraldez lab. Ribo-seq data suggested that there is no alternative translation initiation site in zebrafish for Satb2 excluding the possibility of different forms of Satb2. Since the coverage of this dataset is low, we also performed PCR using primers spanning exon 8 and 9/10. We obtained a single amplicon in both maternal as well as zygotic stages suggesting that only one isoform is expressed (see below).

2. Fig. 2a and Fig. 4a. The genomic feature of the classification should be shown along with the ChIP-seq and ATAC-seq data, because it is not clear if one category is indeed enriched or just appears proportionally to its abundance in the genome.

Response: We think the Reviewer is pointing towards Figure 3a and Figure 4a. We have used the HOMER package for initial annotation of ChIP-seq as well as ATAC-seq peaks. Core promoter region was defined as +/- 2kb from the transcription start site. Exon, Intron and TTS regions were represented from HOMER analysis. Remaining peaks were classified as intergenic regions. Same classification scheme was followed for each dataset. Peak annotation is represented as a percentage of total regulatory sites. This was crucial for stage dependent comparison than the absolute numbers under each category. Details of genomic feature classification were provided in the 'Methods' section. However, in the revised version of the manuscript, we have also described it briefly in the figure legends.

3. Fig. 2e. I understand that the blue circle indicates Satb2 bound genes. Why the numbers differ in the two (left and right for upregulated and downregulated genes in the KO, respectively)?

Response: We think the reviewer is referring to Figure 3e. It is a typographical error which has been fixed in the revised manuscript.

4. Supplementary Fig. 3, 5, and 7. Western blotting panels require the positions of size standards.

Response: We have added size markers to all the western blotting panels as per the suggestion from the reviewer.

Reviewers' Comments:

Reviewer #1:

Remarks to the Author:

The authors have provided extensive and detailed comments/explanations in response to reviewer comments and have modified the manuscript to address the vast majority of the concerns. I am impressed by the revisions and attention paid to the original comments and think the paper will now add important new data that is placed in a broad context of SATB2 function in vertebrates. The extensive scRNAseq data are especially relevant and noteworthy.

Reviewer #3:

Remarks to the Author:

The revised version successfully addressed the previously raised concerns.

Reviewer response NCOMMS-20-39460-T

Reviewer #1 (Remarks to the Author):

The article by Pradhan and colleagues investigates temporal differences in Satb2 function during zebrafish development. The authors focus on early roles for Satb2 that would likely be performed by maternally contributed Satb2 versus later (zygotic) roles for Satb2. They present a large number of sequencing datasets which they use to argue that Satb2 has two distinct roles- one as a repressor in early development, and as an activator later in development. They also suggest that Satb2 may somehow be involved in zygotic genome activation.

A positive of the work described in the manuscript is the amount of sequencing data presented. However, the conclusions drawn from this data are not always consistent within the present analysis and also diverge with what has been shown in other systems (mostly mouse) about Satb2. The primary conclusion at the end of the results, that the contrasting stage-dependent function of Satb2 is a consequence of qualitatively distinct binding, has already been shown in mouse- further it is not stage-dependent, but rather cell-type dependent. This key point is related to a first weakness of the manuscript, which is that the authors have chosen to perform this study in zebrafish but have not utilized key advantages of this model system, such as fluorescent reporters that would allow them to sort specific cell types and really refine cell type specific behaviour of satb2. Such data could provide an important advance in our knowledge of satb2. Instead, they have done large scale tissue evaluations of satb2 function which have the effect of averaging or blending the role of satb2 in these cells.

Response: We thank the reviewer for the thoughtful critique. Indeed, cell type specificity and organismal context are two important factors that need to be considered while interpreting our data and we have tried to do so in the revised version.

Briefly, we sought to use zebrafish as a model as it allowed us to probe the function of Satb2 throughout the developmental progression, from a single cell to an adult. From the outset, we hoped to uncover either completely novel or context-specific functions of Satb2 and how it is deployed temporally to achieve distinct outcomes. As the reviewer pointed out, we also hoped to uncover its cell-type specific activities. We believe that the analysis presented in this manuscript has successfully touched upon most of these issues.

We were keen to glean evolutionary insights using zebrafish as an early vertebrate model that mimics Satb2 associated craniofacial abnormalities first reported in higher

mammals. This is important since Satb1 and Satb2 likely diverged with the evolution of jawed fish. In this context, it is also noteworthy that defined cell types emerge only after early cleavage stages of zebrafish embryogenesis, and thus, we do not attribute the early embryonic function of Satb2 to any specific cell type. This aspect may be unique to zebrafish Satb2 and may differ as opposed to the mouse version of the protein making a direct comparison untenable. We thank the reviewer for raising this important issue and we have tried to account for this distinction in the revised manuscript.

One important novel conclusion that emerged from our studies concerns the functional distinction i.e., bimodality between the maternally deposited and zygotically synthesized Satb2. As described in detail, this bi-modal behavior is pertinent to the early stages of embryonic development.

As pointed out by the reviewer, during later stages, Satb2 likely exhibits cell type specific functions. As alluded to by the reviewer, we considered performing functional studies in the background of a transgenic reporter line. However, due to the nature of the Satb2 mutation, we were unable to obtain homozygous *satb2*^{-/-} fish. Owing to this technical challenge, we could not identify homozygous mutants before subjecting them to FACS sorting necessary to perform the cell type specific analysis. Consequently, throughout the manuscript, we have used embryos obtained by crossing heterozygous parents. The genetic markers which we have characterized in the absence of Satb2 are specific to early neurogenesis and neural crest development (confirmed by previous studies reported in literature and using gene ontology analysis).

To substantiate our conclusions, we have now performed single-cell gene expression analysis in wild type vs mutant embryos. This has allowed us to identify cell types that show significant changes upon Satb2 deletion. Consistent with our previous analysis, scRNAseq data shows maximum reduction in neural progenitors and neural crest cell populations arguing for a tissue specific function. We also observe an increase in the endothelial population and lateral plate mesoderm population. However, we only analyzed early stages of embryos and thus, are not able to comment on effects of loss of function of Satb2 on neural sub-types and future studies will focus on analysis of later stage embryos. Importantly, to analyze effect at later stages of neural development and cell-subtypes experiments, mouse models would prove to be highly relevant.

Throughout our current study, our primary focus was to assess the function of Satb2 during early stages of neurogenesis including its onset. As a result, zebrafish was our system of choice to track neural development during the early developmental time window. Nevertheless, it will be valuable to integrate our findings with the conclusions derived from other model systems to evolve a more complete picture of the process with

a deeper understanding of mechanistic underpinnings underlying neural crest specification and craniofacial development.

As suggested by the reviewer, we also utilized Sox10: GFP reporter lines to analyze the neural crest specific function of Satb2. First, we evaluated and compared the expression of Satb2 in cranial neural crest and non-neural crest cells. Expectedly, we observed significant enrichment in expression levels of Satb2 in the cranial neural crest cells. These results corroborate the transcriptome analysis in crestin: GFP cells which serves as a marker for cranial neural crest cells.

Next, we isolated cranial neural crest and non-neural crest cells using Sox10: GFP transgenic fish line by FACS and performed ChIP-seq for Satb2. We observed significant correlation between datasets generated using sorted population and whole embryos tissue level analysis suggesting that during these stages, both approaches provided similar insights and we can use whole embryo approach at earlier stages when reporter expression is below the detection limit of microscope and flow cytometer.

The authors emphasize the maternal-zygotic transition and argue for a role for satb2 in this transition. But it's not clear what data support this role. The authors have simply shown, at best, that satb2 has a cell type specific function in early, cleavage stage embryos. The fact that satb2 needs to be down-regulated as development moves to different stages of differentiation does not mean that it is required for a maternal-zygotic transition unless this is a unique feature of zebrafish development. This point relates to the second weakness of this paper, which is the absence of clear comparisons to data published in the mouse. This is particularly relevant to the role of Satb2 in early development. In mouse, satb2 is expressed in the blastocyst where it positively regulates Nanog (Savarese et al. 2009), in contrast to the specific repressive role the authors argue for satb2 in zebrafish. It is also not clear what such a specific zygotic transition role would mean for mammals where zygotic gene expression begins in one-to-eight-celled embryos. In mouse, satb2's function in early blastocyst cell fate determination occurs throughout this time from one-celled to hundreds of cells (Goolam & Zernicka-Goetz 2017).

Response: We thank the reviewer for this astute insight. Indeed, the onset of ZGA is a unique feature of zebrafish and thus cannot be directly compared to mammalian ZGA with respect to its timing and cellular changes. However, molecular mechanisms which regulate the process of ZGA are highly conserved between zebrafish and humans (Jukam *et al.* Dev Cell 2017; Schulz and Harrison Nat. Rev Genet. 2019). Importantly, few studies suggest that regulation of ZGA by OCT4 is a conserved mechanism

between zebrafish and humans while mice are the outliers in this regard (Lee *et al.* Nature 2017; Leichsenring *et al.* Science 2013; Gao *et al.* Cell 2018).

We appreciate reviewers' concern that downregulation of *satb2* does not necessarily imply that it is required for a maternal-zygotic transition. Indeed, to assess this possibility, we had engineered continuous expression of *Satb2* by performing overexpression experiments and our data show that sustained expression of *Satb2* during early developmental stages impairs activation of zygotic genes significantly. The effect seems specific in nature as retention of *Satb2* did not affect the maternally deposited genes.

It should be noted that we do not comment on any cell type specific activity of *Satb2* during ZGA, as zebrafish embryos contain only two cell types at this stage: enveloping layer (EVL) and deep cells. The single-cell analysis was employed in this context, simply to highlight cell-autonomous mode of regulation by *Satb2* i.e., we did not intend to highlight cell type specific activity of *Satb2*.

However, our scRNAseq data on embryos expressing higher levels of *Satb2* in mosaic manner suggests negative correlation between *Satb2* and zygotic genes. In our view, these data strongly argue that in zebrafish, downregulation of *Satb2* is necessary for efficient progression of ZGA. We cannot comment on zygotic genome activation in mammals vis-a-vis *Satb2* loss during ZGA. It is important to note that Goolam & Zernicka-Goetz 2017 did not specifically categorize global effects on maternal versus zygotic genes upon overexpression of *SATB2*. However, it would be interesting to follow up on this line in the future and we have now discussed this possibility in the revised version of the manuscript (Line numbers 514-524).

Savarese *et al.* (2009) have suggested positive regulation of *Nanog* by *Satb2* in the early blastocyst, However, as the reviewer correctly pointed out our data do not demonstrate that in zebrafish, *Satb2* downregulates *Nanog*. In fact, *Nanog* expression levels are not perturbed upon over-expression of *Satb2*. This could be due to species-specific differences in *Satb2* activities, and we have discussed this in the revised version of the manuscript (Line numbers 544-548). Our data show that, *Satb2* downregulates *Pou5f3* but upregulates *Sox19b*. Together these data show that the effect of *satb2* on pluripotency factors is qualitatively non-uniform and we hypothesize that ZGA is affected due to such homeostatic imbalances resulting from overexpression of *Satb2*.

As suggested by the reviewer, we have included a section in the discussion about comparison between studies in mice and this analysis (line numbers 541-548).

The study cited by the reviewer (Goolam & Zernicka-Goetz, 2017) is focused primarily on Satb1, another member of the protein family and as such cannot be used for a direct comparison. In fact, these authors mention that Satb2 knockdown does not influence embryonic development adversely presumably due to activity of residual Nanog expression. However, it should be noted that knockdown of Satb2 partly restores gene expression programs which are perturbed upon the knockdown of Satb1. Thus, highlighting antagonism between mouse homologs of Satb1 and Satb2. We have not observed such antagonism between these two proteins at early stages of zebrafish development. and we have discussed this possible species-specific distinction in the revised version of the manuscript (Line numbers 514-524).

In sum, it will be interesting to elaborate on these studies to analyze the effect of Satb2 in other mammalian systems to assess possible limits of conservation across vertebrate species. However, our data and other studies suggest conserved function for Satb2 in neural crest regulation and craniofacial development.

Other major considerations:

1. In Figure 1E, the authors show survival data for their genotypes used in this study. WT individuals have a very low (25% initial survival rate). Why is the survival rate so low among WT fish, and how can the percentage of surviving animals go up over time? There is something not right with these data, and/or something not right with the animals used in this study.

Response: Data represented in this figure summarizes the observed frequency of animals with the corresponding genotype in a test population. For example, on a given day before day 14, if we screen 48 larvae from the in-cross of *satb2*^{+/-}, we observe 25% WT (~ 12), 50% heterozygous mutants (~ 24) and 25% mutant embryos (~ 12) in a clutch in accordance with the Mendelian ratio. Similarly, on day 21, we do not observe any homozygous mutant surviving in a population (or very rare population which we have analyzed using microCT) and thus percent frequencies of Wild type and heterozygous mutants increased to be around 50%. We have now changed the Y-axis label to Survival rate (percent frequency). We hope this provides desired clarification as per the reviewer's request.

2. The authors suggest that “the presence of both SATB1 and SATB2 is only apparent in jawed fish and later species, suggesting these proteins presumably co-evolved with the evolution of the jaw structure.” Co-evolution implies an interaction, like predator-prey co-evolution. It is not clear what the authors are trying to communicate in this section of

the paper. Further, why would gene duplication after the split with agnathans necessarily imply any association with the jaw as opposed to any other structure that is derived in the stem gnathostome lineage, such as bone or teeth, both of which *satb2* has known developmental roles?

Response: Here, by using the term ‘co-evolution’, we implied that the ancestral form of *Satb* protein has diverged into two distinct proteins- *Satb1* and *Satb2* with the evolution of jawed fish. Previous studies have proposed a specific role for *SATB2* during determination of vertebrate jaw structures including teeth (Fish *et al.*, *Evol dev* 2011; Dobрева *et al.*, *Cell* 2006; Sheehan-Rooney *et al.*, *Dev Dyn* 2013). Therefore, we used the term ‘co-evolution’ to indicate the evolutionary dependence of jaw structures on *SATB2*. However, we agree with the reviewer that this does not mean functional interaction between the two and further evidence is required to argue for co-evolution. Therefore, now we use the specific term ‘divergence’. We also appreciate the point of involvement in bone and teeth development. Neural crest cells play an important role in tooth development and skeletal structures, specifically cranial bone development (Yamazak and Hayashi 2004; Mishina and Snider 2014). We have now modified the text to discuss the possible role of *Satb2* in regulating these tissue types as reported in mammals (line numbers 103-105). However, in the current study, we have used jaw development as one of the paradigms to highlight functional conservation and contribution of *Satb2* during neural crest specification.

3. The authors have broadly placed *Satb2* into two categories- maternal and zygotic in a way that implies that they are different proteins rather than the same protein acting in different cells at different times. This causes several downstream problems. First, the analysis of “zygotic” *satb2* does not take into account what is already known about cell-type specific behavior of *satb2*- it is expressed in multiple cell types during development- neural crest, but importantly only a subset of neural crest, neurons, but again only a subset of neurons, and osteoblasts. The analyses on zygotic *satb2* lump all of these together as if *satb2* is doing the same thing in these cells, interacting with the same partners, etc. This has been shown in a number of studies in mice not to be the case. Even in the case of neurons, *satb2* is doing very different things during cortical development compared to its function in adult neurons. So the idea that zygotic *Satb2* is simply a gene expression activator is very over-simplified. That said, the expression data shown are not controversial data- previous work has shown the involvement of *Satb2* in regulating skeletal genes and some NC genes. However, their conclusion that *Satb2* is required for early stages of NC development is not substantiated since they can’t isolate this cell type in their analysis. Also, if this is true, it would be specific to zebrafish since *Satb2* is not expressed in pre-migratory NC in mice. This key difference between zebrafish and mammals is not discussed.

Response: This was perhaps erroneously represented in the earlier version of the manuscript. We did not intend to suggest that maternal and zygotic *satb2* are different protein isoforms. The distinction was only meant to be temporal.

1. maternally deposited protein pool, and 2. zygotically synthesized pool.

We support this by showing that *Satb2* mRNA levels decrease significantly after the onset of ZGA. This suggests that maternally deposited *Satb2* mRNA is depleted from the system. To establish this unequivocally we show that only one form of *Satb2* is expressed throughout early embryogenesis and rule out the possibility of alternate splicing.

We appreciate the point of the reviewer that our analysis on bulk embryos could miss out on the information about cell type-specific regulation by *Satb2* as shown during mouse brain development. However, the purpose of these experiments is to probe how early during the developing embryo, genes that are responsible for neural crest development are positively regulated by *Satb2*. This analysis has helped us identify the critical time window of functional regulation by *Satb2* crucial to facilitate craniofacial development.

Ideally, as the reviewer suggested, this analysis should be performed on isolated neural crest cells from the *satb2* mutants. This is, however, not possible due to the nature of *Satb2* mutation, as homozygous *Satb2*^{-/-} fish can't be easily retrieved.

To circumvent this problem, in the current version of the manuscript, we have included analysis using scRNAseq in WT and mutants. We show specific effect on neural progenitors and neural crest cells (Main Fig. 2). Moreover, *Satb2* ChIP-seq analysis on *Sox10*pos cells also highlights neural crest-specific activity of *Satb2* (Main Fig. 3). We are sensitive to the fact that these data do not allow us to probe into the role of *satb2* during specification of neuronal subtypes. Future studies will be necessary to address this question.

A recent study by Fazio *et al.* (Biorxiv <https://doi.org/10.1101/2020.11.01.364406>) reported a knockdown with morpholinos against *Satb2*. Similar KD approach was used by Ahn *et al.* 2010 to show that loss of *Satb2* results in reduced *sox10* expression (assessed by *sox10:gfp* reporter line) and craniofacial defects. However, it does not affect melanocyte development. Together these data suggest that *Satb2* has a specific function in cranial neural crest cells and not in neural crest cells which contribute to peripheral tissue types.

Figure 1. Data from Fazio et al. 2020 suggesting loss of function of Satb2 using previously characterized morpholinos leads to defective cranial skeleton (marked by Sox10:mCherry) but does not affect melanin pigmentation.

We also do not see the loss of pigmentation in the survivor Satb2 mutants (Supplementary Fig1) essentially supporting conclusions from this study. Moreover, in our manuscript, we address this question and subsequently support this notion by showing specific enrichment of Satb2 in head tissue of developing mouse embryos at (E9.5). We show higher enrichment in case of head tissue (source of cranial neural crest cells than trunk tissue). New analysis in Sox10^{pos} cells also suggest higher enrichment in cranial neural crest cells than non-neural crest cells in zebrafish.

Comments from the reviewer further suggest that Satb2 function in early neural progenitors could be restricted to zebrafish since Satb2 is not expressed in murine neural crest progenitors. However, RNA-seq studies from the laboratory of Filippo Rijj (Minoux *et al.* Science 2017) suggests significant expression of Satb2 in neural crest progenitors at neural crest subtypes from as early as E8.5.

Figure 2. Bar plot representing SATB2 expression in isolated neural progenitors and neural crest subtypes from developing mouse embryos.

Moreover, scRNAseq analysis of cranial neural crest cells (E8.5) and trunk neural crest cells (E9.5) also suggest enrichment of *Satb2* in cranial neural crest population compared to neural crest contributing to peripheral tissues (Soldatov *et al.* Science, 2018). This data also supports our ChIP-seq studies in mouse.

Figure 3. Expression of *Satb2* from scRNAseq dataset (Red: positive to blue: negative expression) <http://pklab.med.harvard.edu/ruslan/neural.crest.html>

4. on page 10, the authors suggest "that unlike zygotic *Satb2*, the maternal pool of *Satb2* is involved in transcriptional repression and possibly regulates the same group of genes in an opposite manner." They have the chip-seq data which could address this claim. They do use this data later to argue that *satb2* is binding different regions maternally vs zygotically (fig. 7a). So, both of these statements can't be true.

Response: In Figure 7A, we demonstrate that Satb2 binds in a differential manner at distinct stages (intergenic versus promoter regions). However, the nearest genes are the same. This is an interesting point and we have included this in the results/discussion sections (line numbers) suggesting not only differential interacting partners but also binding at different genomic regions potentially contributing to Satb2's contrasting functions. For example, we observe that Satb2 binds only intergenic regions of neural crest marker genes such as *sox10* at 512 and dome stage. This pattern of occupancy changes during later stages as Satb2 also binds the promoter region of *sox10*.

4. The authors employ gene expression analyses at 80% epiboly, 5-6 ss and 14 ss, in order to study NC development. First, it has previously been shown that Satb2 is widely, if not ubiquitously, expressed in the early zebrafish embryo (Ahn et al. 2010), so these data do not specifically address NC. As noted above, it is not clear why lineage markers would not be used to provide more specific analyses of cell type behavior- the authors appear to assume that *satb2* is doing the same thing in all cells in this zygotic period.

Response: Ahn *et al.* performed 'in situ' analysis which did not include stages of neural crest specification (6 somites) and NC differentiation (14 somites) in their analysis. In contrast, quantitative RNAseq analysis has provided detailed insights into the expression pattern of Satb2.

Fazio *et al.* (Biorxiv <https://doi.org/10.1101/2020.11.01.364406>) also analyzed previously published data to show that *satb2* is highly expressed in a migrating *crestin:EGFP+* NC cells sorted from zebrafish embryos at the 15 somite stage (Kaufman et al., 2016). As mentioned earlier, the dataset from Ahn *et al.* (2010) does not include this stage. Based on the RNA seq study performed by Kaufman *et al.*, it is clear that during early stages of neural crest development, Satb2 plays a cell type (NC) specific function.

**Zebrafish Neural Crest Tg(*crestin:EGFP*) sorted cells
(Kaufman et al. 2016)**

	crestin - (FPKM)	crestin + (FPKM)	Log2 Fold Change
satb2	0.775	13.597	4.134

(Fazio et al. 2020)

Figure 4. FPKM values for Satb2 in neural crest and non-neural crest cells (Kaufman et al. 2016).

We have validated observations from RNA-seq studies using qPCR utilizing another reporter line for marking neural crest population- Sox10: GFP. We isolated RNA from Sox10+ and Sox10- cells at 14 somites stage and observed that *Satb2* is indeed enriched in the neural crest population than in the non-neural crest thus further strengthening our conclusions (Supplementary Fig. 2a).

5. Further, and more importantly, the expression data reported by Ahn *et al.* 2010 are not consistent with the biphasic model of *satb2* expression reported here. The authors do not discuss this point.

Response: We thank the reviewer for pointing out this oversight and we have now discussed this point in the revised manuscript.

It should be noted that the expression data provided by Ahn *et al.* 2010 is non-quantitative WISH analysis. We have used high depth stage-dependent transcriptome data which allowed us to confirm and extend the biphasic nature of *Satb2* during early embryonic development. We analyzed datasets from two independent studies (Pauli *et al.*, Genome Res 2012 and EBI). Moreover, our analysis of the human stem cell differentiation model also supports this notion. This point is also discussed in the revised version (Line number 483-485).

Minor considerations:

Fig. 1, panel C- why does mutant band run lower in KO vs het?

Response: We appreciate this concern. This anomaly could be due to specific issues with gel electrophoresis run. We have now replaced this panel with a new plot. High-resolution melt curve analysis confirms that the product size is the same in heterozygous and homozygous mutants (assessed by T_m) confirming our conclusions.

Reviewer #3 (Remarks to the Author):

In this manuscript, Pradhan et al. analyzed Satb2 localization and function in zebrafish and other model systems. They first investigated the knockout phenotype in zebrafish and found craniofacial defects as observed in mammals, implying that analyzing the Satb2 function in zebrafish is relevant to human disease. They then went into genome-wide analysis, including, RNA-seq, ChIP-seq, and ATAC-seq, which suggested that the zygotic knockout phenotype results from dysregulation of gene expression through altered chromatin structure. The aberrant expression of some key genes was confirmed by in situ hybridizations. To analyze the function of maternal Satb2, its overexpression was employed. The overexpression induced chromatin silencing rather than activation. The differential effects of Satb2 in the zygotic genome activation and in differentiation may be explained by the partner transcription factor. The function of biphasic expression of Satb2 was thus explained.

The study was well performed and thorough, providing significant advance in understanding Satb2 function during development and differentiation. Furthermore, this study provokes our thinking about how a chromatin protein can bimodally regulate gene expression. The conclusions were well-supported by the high-quality data presented.

I have a few comments that may improve the manuscript.

1. There is no consideration for the splice variants of Satb2, while the existence of variants is suggested when generating the phylogenetic tree. Could the authors exclude a possibility that maternal and zygotic Satb2 protein differ in function due to alternative splicing and/or alternative translation initiation? This question can be experimentally addressed, but at least, the authors should look into the database more and comment on it.

Response: During phylogenetic tree construction, we excluded the splice variants of SATB1 and we did not find any splice variants for SATB2. Additionally, RNA sequencing data also confirms the absence of detectable splice variants for SATB2 in Zebrafish. To improve the clarity, the methodology section has been modified. We have specifically included the following sentence to clarify this point *“SATB1 splice variants were removed, and no splice variants were observed for SATB2.”*

To further validate that there are no splice variants expressed during these stages of zebrafish development, we analyzed Ribo-Seq data from Giraldez lab. Ribo-seq data suggested that there is no alternative translation initiation site in zebrafish for Satb2 excluding the possibility of different forms of Satb2. Since the coverage of this dataset is low, we also performed PCR using primers spanning exon 8 and 9/10. We obtained a single amplicon in both maternal as well as zygotic stages suggesting that only one isoform is expressed (see below).

2. Fig. 2a and Fig. 4a. The genomic feature of the classification should be shown along with the ChIP-seq and ATAC-seq data, because it is not clear if one category is indeed enriched or just appears proportionally to its abundance in the genome.

Response: We think the Reviewer is pointing towards Figure 3a and Figure 4a. We have used the HOMER package for initial annotation of ChIP-seq as well as ATAC-seq peaks. Core promoter region was defined as +/- 2kb from the transcription start site. Exon, Intron and TTS regions were represented from HOMER analysis. Remaining peaks were classified as intergenic regions. Same classification scheme was followed for each dataset. Peak annotation is represented as a percentage of total regulatory sites. This was crucial for stage dependent comparison than the absolute numbers under each category. Details of genomic feature classification were provided in the 'Methods' section. However, in the revised version of the manuscript, we have also described it briefly in the figure legends.

3. Fig. 2e. I understand that the blue circle indicates Satb2 bound genes. Why the numbers differ in the two (left and right for upregulated and downregulated genes in the KO, respectively)?

Response: We think the reviewer is referring to Figure 3e. It is a typographical error which has been fixed in the revised manuscript.

4. Supplementary Fig. 3, 5, and 7. Western blotting panels require the positions of size standards.

Response: We have added size markers to all the western blotting panels as per the suggestion from the reviewer.